# Small molecule splicing modifiers with systemic HTT-lowering activity

Anuradha Bhattacharyya[1], Christopher R. Trotta [1], Jana Narasimhan[1], Kari J. Wiedinger[1], Wencheng Li[1], Kerstin A. Effenberger[1], Matthew G. Woll[1], Minakshi B. Jani[1], Nicole Risher[1], Shirley Yeh[1], Yaofeng Cheng[1], Nadiya Sydorenko[1], Young-Choon Moon[1], Gary M. Karp[1], Marla Weetall[1], Amal Dakka[1], Vijayalakshmi Gabbeta[1], Nikolai A. Naryshkin[1], Jason D. Graci[1], Thomas Tripodi Jr.[1], Amber Southwell[2], Michael Hayden[3], Joseph M. Colacino[1] & Stuart W. Peltz [1]✉

Huntington's disease (HD) is a hereditary neurodegenerative disorder caused by expansion of cytosine-adenine-guanine (CAG) trinucleotide repeats in the huntingtin (*HTT*) gene. Consequently, the mutant protein is ubiquitously expressed and drives pathogenesis of HD through a toxic gain-of-function mechanism. Animal models of HD have demonstrated that reducing huntingtin (HTT) protein levels alleviates motor and neuropathological abnormalities. Investigational drugs aim to reduce HTT levels by repressing *HTT* transcription, stability or translation. These drugs require invasive procedures to reach the central nervous system (CNS) and do not achieve broad CNS distribution. Here, we describe the identification of orally bioavailable small molecules with broad distribution throughout the CNS, which lower *HTT* expression consistently throughout the CNS and periphery through selective modulation of pre-messenger RNA splicing. These compounds act by promoting the inclusion of a pseudoexon containing a premature termination codon (stop-codon psiExon), leading to *HTT* mRNA degradation and reduction of HTT levels.

[1] PTC Therapeutics, Inc. 100 Corporate Court, South Plainfield, NJ, USA. [2] Burnett School of Biomedical Sciences, College of Medicine, University of Central Florida, Orlando, FL, USA. [3] Centre for Molecular Medicine and Therapeutics, Department of Medical Genetics, University of British Columbia, Vancouver, BC, Canada. ✉email: speltz@ptcbio.com

Huntington's disease (HD) is an autosomal dominant progressive neurodegenerative disorder. HD is caused by cytosine–adenine–guanine (CAG) repeat expansions in the huntingtin (*HTT*) gene resulting in the production of a ubiquitously expressed pathogenic mutant HTT (mHTT) protein[1–3]. HD is characterised by progressive atrophy of the striatum, cortex and other areas of the brain that causes motor, cognitive and psychiatric symptoms. Neurodegeneration progresses to the thalamus, substantia nigra pars reticulata and subthalamic nucleus in advanced disease[4–6].

Currently, there are no approved disease-modifying treatments for HD. However, significant advancements are being made in identifying huntingtin (HTT) protein-lowering therapies using multiple approaches, including ribonucleic acid (RNA) interference using short interfering RNAs, short-hairpin RNAs, or microRNAs and antisense oligonucleotides causing translational repression or messenger RNA (mRNA) degradation. Transcriptional repression approaches using zinc finger proteins or clustered regularly interspaced short palindromic repeats (CRISPR)/CRISPR-associated protein 9 (Cas9) have also been employed[1,2].

The key challenge for these HTT-lowering therapies is optimal delivery and distribution throughout the central nervous system (CNS). These agents do not cross the blood–brain barrier, they require intrathecal or intraparenchymal delivery, and result in uneven CNS distribution and inconsistent HTT lowering throughout the CNS. Peripheral morbidities such as weight loss and skeletal muscle atrophy can occur and may contribute to disease severity[7,8]. Therefore, an orally bioavailable, systemically distributed, brain-penetrating small molecule that reduces the toxic burden of mHTT protein uniformly throughout the brain and peripheral tissues would be highly beneficial for HD management.

We have developed a drug discovery splicing platform to identify compounds that modulate splicing. This platform can be used to develop target-selective small molecules, such as the first-ever orally bioavailable modifier of survival of motor neuron 2 (*SMN2*) splicing for the treatment of spinal muscular atrophy (SMA)[9]. Evrysdi™ (risdiplam) was discovered using this platform and has recently been approved by the Food and Drug Administration and several additional regulatory authorities[10]. Here, we describe the discovery of a class of small molecule splicing modifiers that were specifically synthesised to promote selective splicing of an inducible pseudoexon containing a premature termination codon (stop-codon psiExon), reducing huntingtin mRNA and protein levels in cells and animal models. These results extend the application of our splicing platform to include the identification of orally bioavailable compounds to treat diseases by decreasing gene expression.

## Results

**Identification of HTT-lowering compounds**. HTT-lowering compounds were identified by screening ~300,000 diverse molecules from a proprietary chemical library. The screening was performed using a highly sensitive and robust HTT protein detection assay in fibroblasts isolated from HD patients (HD fibroblasts). Multiple classes of active compounds (hits) were identified, including heat shock protein 90 inhibitors (HTT-A) previously shown to reduce mHTT levels[11] (Supplementary Fig. 1a–c) and splicing modifiers HTT-C1 and HTT-D1 (Fig. 1a), two compounds similar to molecules identified in the SMA drug development programme[12,13].

HD fibroblasts were treated with the splicing modifiers, and m*HTT* mRNA and protein levels were determined using reverse transcription-quantitative polymerase chain reaction (RT-qPCR), electrochemiluminescence (ECL) and western blot analysis. Both HTT-C1 and HTT-D1 dose-dependently decreased m*HTT* mRNA and protein levels (Fig. 1b–d; Fig. 2a; Supplementary Fig. 1d). Since most HD patients carry both wild-type (wt) and m*HTT* alleles, we quantified wt HTT protein levels and demonstrated similar decreases in the control fibroblasts (Supplementary Fig. 1e).

**Compounds affect *HTT* pre-mRNA splicing**. *HTT* pre-mRNA splice junctions were characterised using primer walking. Differential splicing was observed between exons 49 and 54 (Fig. 2b, Supplementary Fig. 2a, b). Splice junctions from these samples were also analysed using targeted next-generation sequencing (NGS; AmpliSeq™). The combined results demonstrated that the compounds induced inclusion of an exon derived from a sequence inside intron 49 of the *HTT* pre-mRNA (Fig. 2c–e; junction expression index (JEI) reduction of >25%; Supplementary Fig. 2c). This intronic sequence represents a compound-inducible pseudoexon (psiExon) which, when included, alters the coding sequence and introduces a premature translation termination codon, leading to the degradation of *HTT* mRNA through nonsense-mediated mRNA decay (NMD) (Fig. 2f), and reduction in HTT protein levels.

Analysis of this stop-codon psiExon shows that it has weak 5′ and 3′ splice sites (ss) (Supplementary Fig. 3). The 5′ss has a noncanonical GA dinucleotide at the −2 and −1 positions, differing from the canonical AG sequence. Splicing of exons with a GAgu 5′ss is inefficient (as is the case with *SMN2* exon 7[14]) and can be enhanced using splicing modifiers such as risdiplam[9,15,16] and other investigational compounds[13,17,18]. However, the inclusion of psiExons by these molecules had not been previously reported[12,13,17].

**Global effects of splicing modification**. To determine the full extent of gene expression (mRNA-level) and splicing (exon-level) changes by HTT-lowering splicing modifiers, we performed RNA-Seq analysis in human SH-SY5Y cells treated with HTT-C2 (branaplam; a more potent, but structurally similar analogue of HTT-C1), or control (dimethyl sulphoxide; DMSO) (Supplementary Fig. 4a). At 24 nM (twice the inhibitory concentration$_{50}$ [$IC_{50}$] of 12 nM), HTT-C2 downregulated the expression of wt *HTT* and a few other genes (Fig. 3a). After treatment with 100 nM HTT-C2, a concentration eightfold the $IC_{50}$, additional mRNAs were downregulated, indicating a dose-dependent effect and a favourable selectivity window for wt *HTT* over other genes at the lower concentration (24 nM) (Supplementary data 1).

Splicing analysis demonstrated that, in addition to the *HTT* stop-codon psiExon49a inclusion, 165 and 215 splicing events were altered when treated with 24 nM and 100 nM HTT-C2, respectively (Supplementary data 2). The majority of splicing events resulted in inclusion (Inc) or skipping (Skp) of cassette exons (CE; exons either included or skipped to create distinct protein isoforms[19]) (Fig. 3b, c). Inc events were more common (1.4×–3.3×) than Skp events (Fig. 3d). Similar to *HTT* stop-codon psiExon49a, many of the Inc CEs (22% and 44% in 24 nM and 100 nM HTT-C2, respectively), had no annotations for at least one ss (Fig. 3e; Supplementary Fig. 4b), and had significant enrichment of −2, −1 GA, and −3 A, +3 A sequence at the 5′ss (represented by AGAguaag) of exons activated by HTT-C2 (Fig. 4a, b, Supplementary data 2). These 31 psiExons are minimally included at basal level (Fig. 3f; median percent-spliced-in [PSI] index = 0.7%), poorly conserved (Fig. 3g), shorter (median size = 64 bps), and with significantly weaker 5′ss than annotated exons (Supplementary Fig. 5). Inclusion of these psiExons containing a stop codon or causing frameshift (NMD-psiExons) were correlated with downregulation of their host

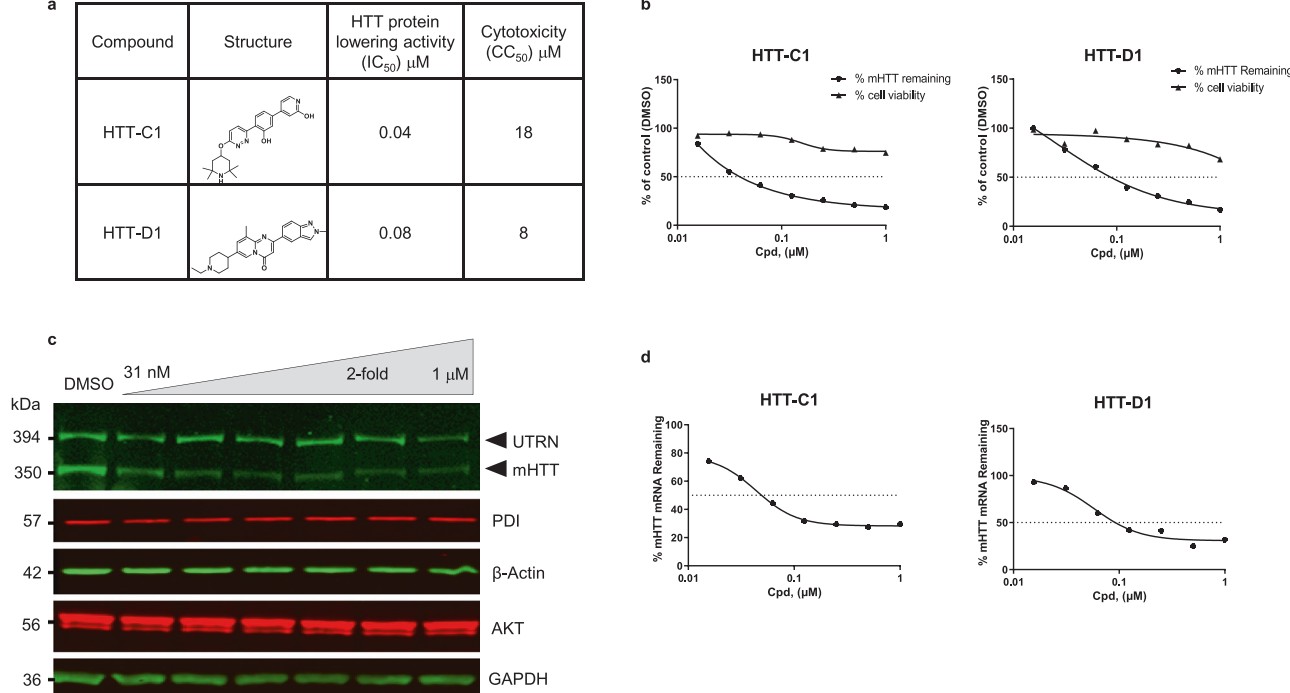

**Fig. 1 Huntingtin (HTT)-lowering activity in vitro. a** Chemical structures of HTT-C1 and HTT-D1. **b** Electrochemiluminescence (ECL) analysis of mutant HTT protein from fibroblasts isolated from a homozygous patient with Huntington's disease (HD) (GM04857) after 96 h of continuous treatment with HTT-C1 and HTT-D1 (0.01–1.0 μM). Representative graphs show percent HTT remaining relative to the dimethyl sulphoxide (DMSO) control. Cell viability assays were performed in parallel. Data represent mean of two ($n = 2$) biologically independent samples per data point from one dose–response experiment. **c** Western blot of HTT protein and housekeeping proteins, oxidoreductase-protein disulphide isomerase (PDI), beta-actin, alpha serine/threonine-protein kinase (AKT) and glyceraldehyde-3-phosphate dehydrogenase (GAPDH) in HD fibroblasts after 96 h of continuous treatment with HTT-C1 (0.015–1.0 μM). Utrophin (UTRN) was also used as a loading control. The western blot data used a representative splicing modifier (tested at multiple concentrations) from a single experiment. Multiple splicing modifiers from the same class were tested and evaluated by western blot analyses. **d** Reverse transcription-quantitative polymerase chain reaction (RT-qPCR) analysis of *HTT* mRNA in patient fibroblasts after 24 h of treatment with HTT-C1 and HTT-D1 (0.01–1.0 μM). Representative graphs show percent *HTT* mRNA remaining relative to DMSO control; normalised to housekeeping gene, TATA-box binding protein. Data represent mean of two ($n = 2$) biologically independent samples per data point from one dose–response experiment.

genes ($P < 0.05$, Fig. 3h, Supplementary data 1), suggesting an NMD mechanism similar to *HTT*.

The strong preference for AGAguuag 5′ ss for HTT-lowering splicing modifiers prompted us to compare the effect of the *SMN* splicing modifier SMN-C3 (a close analogue of risdiplam) on mHTT lowering and global splicing changes. We observed that SMN-C3 demonstrated a striking 170-fold decrement in potency for mHTT as compared to HTT-C2, whereas the molecules show similar SMN EC$_{1.5X}$ (Supplementary Fig. 6). When splicing effects were analysed globally by examination of previously published RNA-Seq data[12], treatment with SMN-C3 led to the generation of 33 psiExons with the predominant consensus of ADGAguaag (D = U, A or G) (Fig. 4a, b, Supplementary data 2), consistent with the inclusion of *SMN* exon 7 with an AGGAguaag 5′ ss. Together, these data demonstrate that HTT-C2 and SMN-C3 represent two distinct classes of splicing modifiers that target 5′ ss with the noncanonical GA dinucleotide at position −2, −1 of the 5′ ss, with the distinction being a preference for adenosine at either −3 for *HTT*-selective molecules or −4 for *SMN*-selective molecules.

The strong structure-based sequence selectivity between the two classes of splicing modifiers for *HTT* and *SMN* splicing supports previous data proposing that these molecules bind a specific RNA interface created by the noncanonical GAgu 5′ ss in complex with U1[13,18]. To further validate that the stabilisation of the U1/pre-mRNA complex leads to the induced inclusion of the psiExons identified through our small molecule treatment, we transfected human embryonic kidney 293 (HEK293) cells with a

variant U1 small nuclear RNA (snRNA) (U1-GA variant). This variant is re-programmed to perfectly bind to a noncanonical GAgu 5′ss and was shown to activate the wt *HTT* psiExon49a and 23 other psiExons activated by 100 nM HTT-C2 (Supplementary Fig. 7a, b). Interestingly, 582 additional psiExons were activated only by the U1-GA variant (Supplementary Fig. 7b, Supplementary data 3) and were enriched for GA at the −2 to −1 position of 5′ss, with no preference for adenosine at the −4 or −3 positions (Supplementary Fig. 7a). These results further support the hypothesis that HTT-C2 functions to stabilise U1-5′ss interaction of a highly selectively subset (AGAguaag 5′ss) of potential psiExons located throughout the human genome, although further experiments are needed to prove a direct interaction.

**Elements required for psiExon inclusion.** The importance of the AGAguaag 5′ss sequence of *HTT* stop-codon psiExon49a and the species-specific activity of the compounds was confirmed in a series of minigene experiments using human and mouse wt *HTT* sequences, where alterations to the −2 and −1 sequences demonstrated requirements for GA and −2, −1 position (Fig. 4c, d, Supplementary Fig. 8a–d).

Although the RNA-Seq data identified a limited number of psiExon inclusion events occurring by either direct or compound-induced U1 recruitment ($n = 609$; Supplementary Fig. 7b), we developed a bioinformatic algorithm (see Methods) and identified >58,000 potential psiExons with an AGAguaag 5′ss throughout the human genome, including four within introns 1, 8 and 40 of

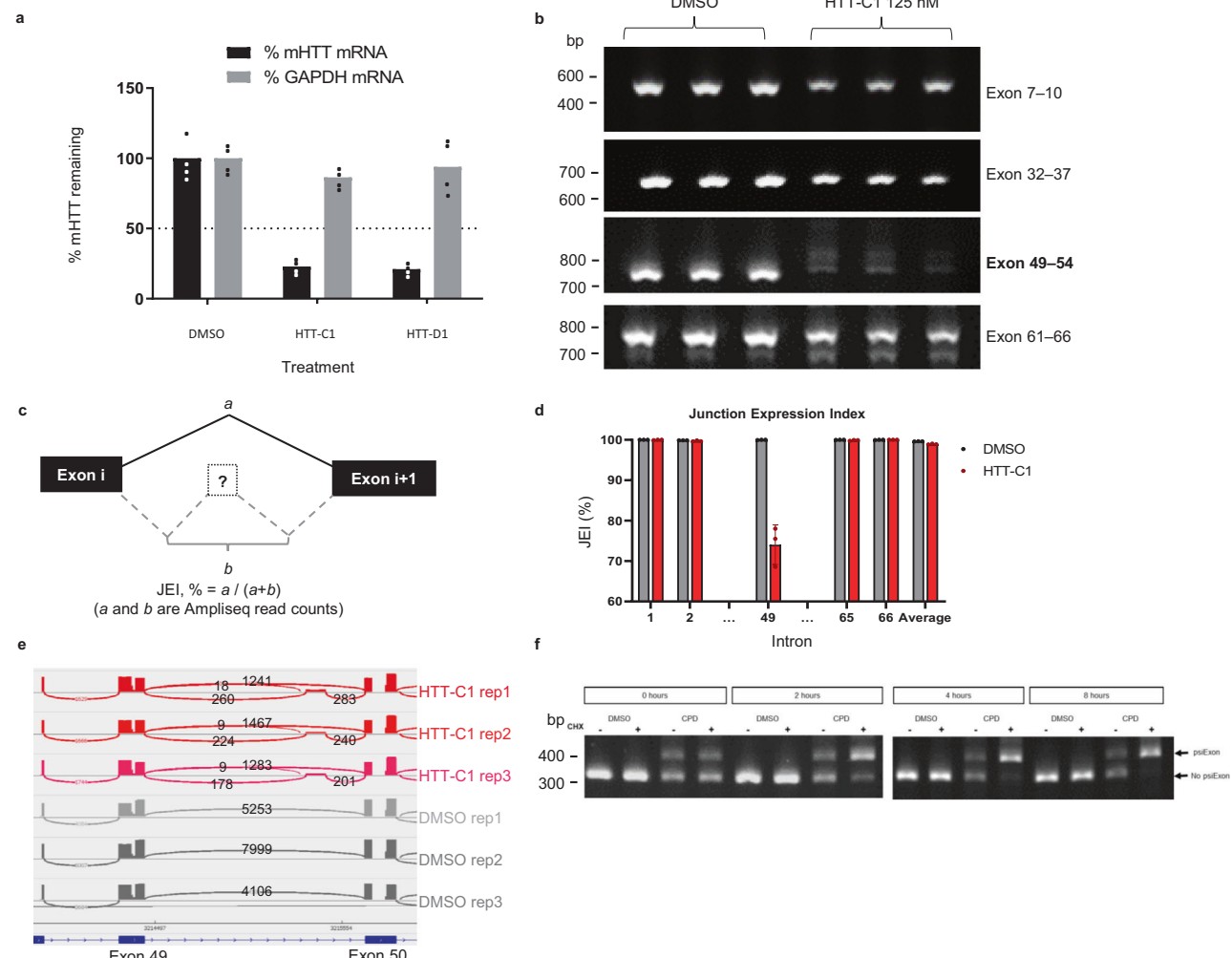

**Fig. 2 Splicing of human huntingtin (_HTT_) pre-mRNA resulting in lowering of _HTT_ mRNA. a** Reverse transcription-quantitative polymerase chain reaction (RT-qPCR) analysis of _HTT_ mRNA in B-lymphocytes from the same patient (GM04856 cells) after 24 h of treatment with HTT-C1 and HTT-D1 (0.25 μM). Representative graphs show percent _HTT_ mRNA remaining relative to dimethyl sulphoxide (DMSO) control; normalised to housekeeping gene, glyceraldehyde-3-phosphate dehydrogenase (_GAPDH_). Data represent the mean of two ($n = 2$) biologically independent samples per data point. **b** Reverse transcription-polymerase chain reaction (RT-PCR) analysis of _HTT_ mRNA after 24-hour treatment with 125 nM HTT-C1 or DMSO in patient-derived B-lymphocytes (GM04856). The data are from a single experiment with three biologically independent samples per data point. The comprehensive data set is provided in Supplementary Fig. 2. **c** Diagram illustrating how the junction expression index (JEI) was calculated for each of the 66 introns in the _HTT_ gene. **d** JEI of intron 49 and a selection of other introns. The JEI of intron 49 was significantly reduced, indicating a splicing event (>25% reduction; $P < 0.05$). Average JEIs are shown as bars. Error bars represent standard deviation. Data were based on three biological replicates of next-generation sequencing (NGS) data. **e** Sashimi plot of alternative splicing (AS) within intron 49 of the _HTT_ pre-mRNA using NGS data. A minimum threshold of five reads was used to visualise these data in the Integrative Genomics Viewer. **f** Endpoint polymerase chain reaction (EP-PCR) analysis of GM04856 cells treated with DMSO or 250 nM HTT-C1. After 18 hrs, cells were treated with 10 μM cycloheximide or DMSO. Total RNA was isolated at 0, 2, 4 and 8 hrs. The data are from a single experiment with two biologically independent samples per data point.

the _HTT_ gene (Supplementary Fig. 8c). The lack of response of these potential psiExons to either HTT-C2 or the U1-GA variant suggests that additional sequence elements are required to promote psiExon inclusion. This is supported by the examination of splicing of minigenes in which the mutation of −2, −1 positions to canonical AG of psiExon49a resulted in its constitutive inclusion, while the same mutation in other potential psiExons had no effect (Supplementary Fig. 8d).

Bioinformatic analysis of _HTT_ stop-codon psiExon49a identified a number of potential sequences that could function as exonic splicing enhancers (ESE) upstream of the 5′ss (Supplementary Fig. 8e). Deletion of 20 nucleotides (−38 to −19), and more precise partial deletion or mutation of nucleotides CAGGA at the −38 to −34 positions resulted in the total loss or reduced

compound-induced splicing (Fig. 4e, Supplementary Fig. 8f, g), suggesting the functional importance of this region.

**HTT-C2 elicits mHTT lowering in HD mice.** As our splicing modifiers require the presence of the human _HTT_ intron 49, we studied the pharmacodynamic effects of HTT-C2 in the BACHD mouse model that expresses a full-length human m_HTT_ gene[20]. Compound HTT-C2 was evaluated due to its superior exposure relative to HTT-C1 (Fig. 5a). Daily oral HTT-C2 reduced mHTT levels within brain tissue of mice in a dose-dependent manner (Fig. 5b). Maximal reductions of mHTT levels were achieved by Day 21 of treatment, with no further reduction observed (Fig. 5c). These effects were reversible, as protein expression levels returned

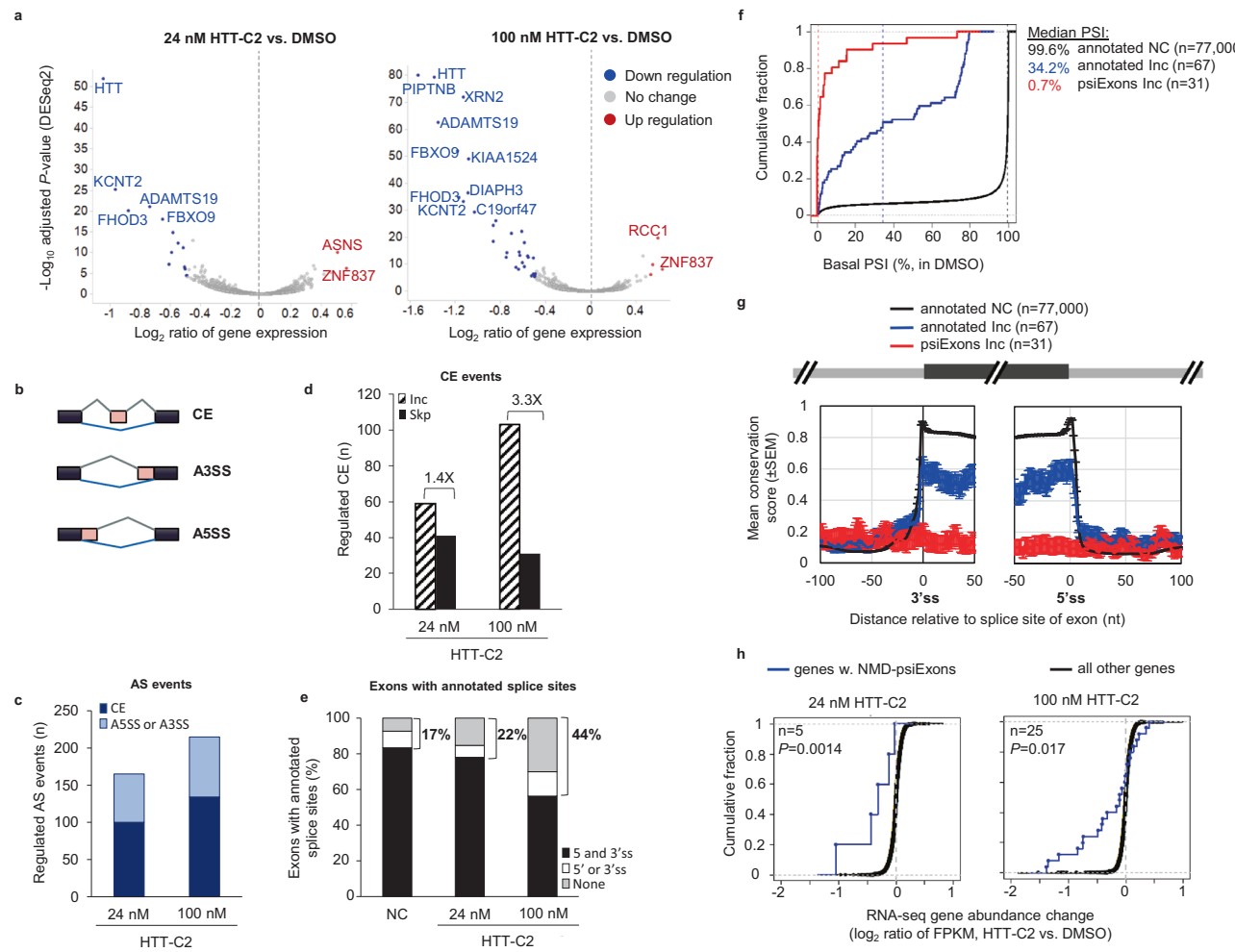

**Fig. 3 Selectivity of compound-induced splicing. a** Volcano plot of RNA-Seq analysis comparing gene expression in SH-SY5Y cells treated with either 24 nM or 100 nM of HTT-C2 with dimethyl sulphoxide (DMSO) treatment. mRNAs with significant changes in expression (>1.5-fold, false discovery rate (FDR) < 5%) are shown as blue and red dots for down- and upregulation, respectively. **b** A schematic of alternative splicing (AS) events. CE, cassette exon; A3SS, alternative 3′ splice sites (ss); A5SS, alternative 5′ss. **c** Number of regulated AS events in SH-SY5Y RNA-Seq data following treatment with 24 nM and 100 nM HTT-C2. **d** Number of CEs inclusion (Inc) or skipping (Skp) after HTT-C2 treatment; ratio of Inc/Skp are shown in text. **e** Percentage of exons with 3′ and 5′ss annotated by public databases (Refseq, Ensembl, or UCSC Known Genes) for NC (no change) or Inc exons. **f** Cumulative distribution function (CDF) curves of basal percent-spliced-in (PSI) index (average PSI in DMSO samples). The graph shows data for exons separated into three groups; Inc is based on ΔPSI > 20% and two-sided Fisher's Exact Test $P < 0.001$ in any one of the two conditions (24 nM or 100 nM HTT-C2 vs. DMSO). Median values are shown as dashed vertical lines for each group. **g** Sequence conservation of 3′ss and 5′ss region. Conservation is based on phastCons score for 46-way placental mammals. Mean (standard error of mean [SEM]) conservation scores are shown. **h** CDF curves of RNA-Seq mRNA abundance change for genes with predicted nonsense-mediated mRNA decay (NMD)-psiExons. NMD-psiExons are psiExons whose inclusion in mRNA introduces a premature termination codon or causes frameshift or both, and are included (Inc) following HTT-C2 treatment. Number of genes (n) and $P$ value are indicated. $P$ value is based on comparison with "all other genes" group using Wilcoxon rank-sum test (two-sided).

to control levels within 10 days of treatment cessation (Fig. 5d). Uniform >50% mHTT protein-level lowering was achieved throughout the brain following HTT-C2 treatment, most importantly within the striatum and cortex (Fig. 5e).

**Optimisation of *HTT* splicing modifiers**. Our splicing modifiers target products of both *HTT* alleles by promoting *HTT* stop-codon psiExon49a inclusion that leads to similar reduction of wt and mHTT levels (Supplementary Fig. 1e). Research suggests that 50% reduction of mHTT would be beneficial to HD patients, and a 50% global reduction of wt HTT would be well tolerated[21,22], which highlights the need for therapies that lower HTT in the brain while not depleting HTT in the periphery. However, HTT-C2 treatment had a far greater mHTT-lowering effect in peripheral tissues (>90%) than in the brain (Supplementary Fig. 9a).

As P-glycoprotein (P-gp) is one major transport protein expressed on BBB, which limits the entry of various drugs into the CNS[23], HTT-C2 and other compounds were tested in an in vitro assay using Madin–Darby canine kidney cells over-expressing human P-gp (Supplementary Fig. 9c). HTT-C2 was determined to be a P-gp substrate. Subsequently, a lower unbound concentration of HTT-C2 was confirmed in the mouse brain compared with plasma by measuring the unbound brain partition coefficient ($K_{p,uu}$) (Fig. 5h). This suggests that reduced mHTT lowering in the brain versus the periphery was most likely attributed to P-gp efflux.

To achieve similar lowering between the periphery and brain, chemical optimisation led to HTT-D3, an equally potent compound with reduced P-gp efflux (Fig. 5h). Oral administration of HTT-D3 resulted in dose-dependent and approximately

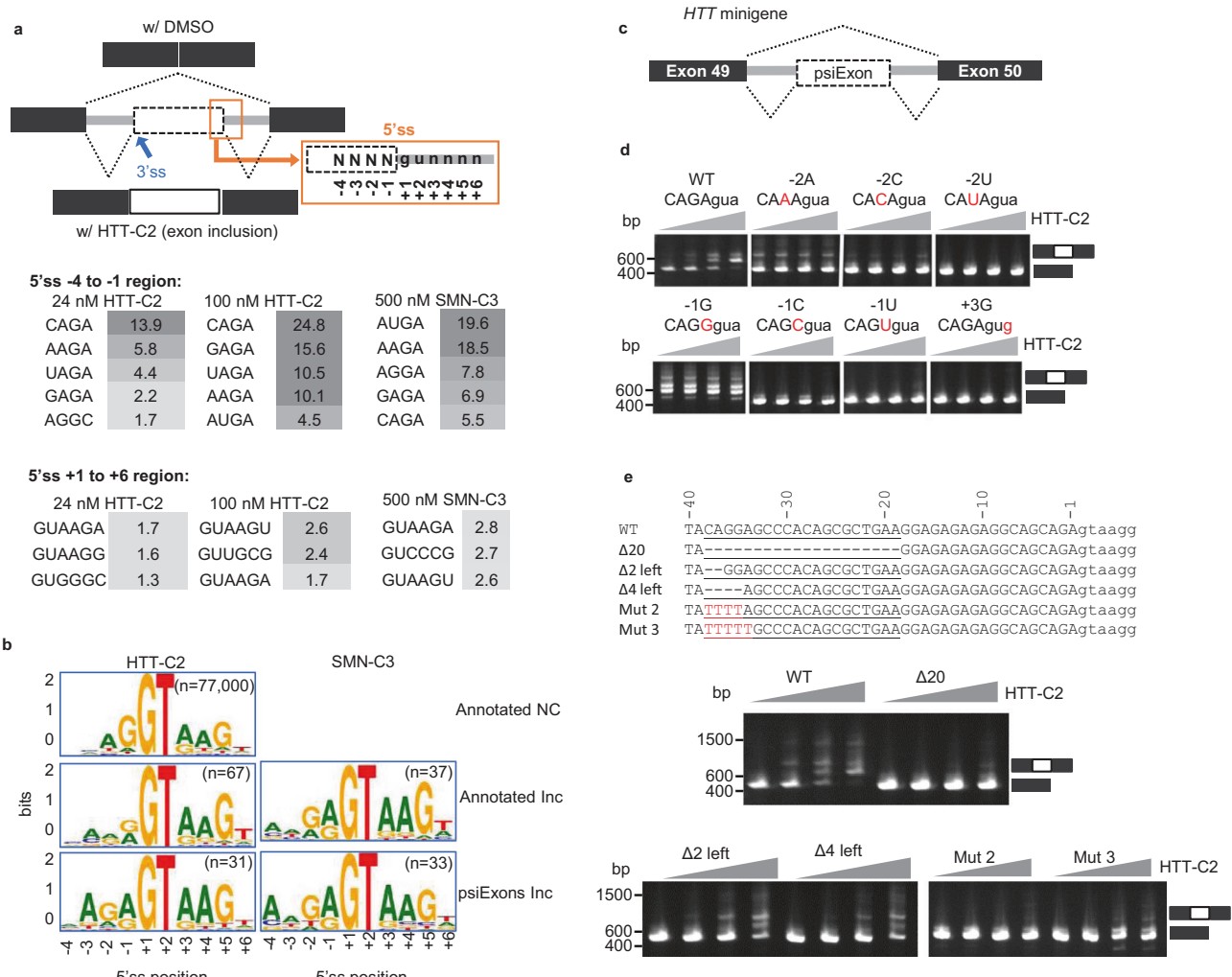

**Fig. 4 Regulation of compound-induced splicing for HTT-C2 and SMN-C3. a** 5′ splice sites (ss) sequence at regions (−4 to −1 and +1 to +6) were studied for the enrichment of Inc (included) vs. NC (no change) exons. Significance scores are shown. Wide boxes represent exons. **b** Schematic of 5′ss sequence logo in the three exon groups: annotated NC, annotated (inclusion) Inc and psiExons Inc. **c** Diagram illustrating the design of huntingtin (*HTT*) minigene constructs for studying compound-induced splicing. **d** Polymerase chain reaction (PCR) analysis of RNA extracts from HEK293 cells transfected with wild-type (wt) human *HTT* minigene or constructs with point mutations in the −2 to +3 region of the 5′ss; cells were treated with dimethyl sulphoxide (DMSO) or HTT-C2 (0.010–1 μM). The data are from a single transfection experiment with multiple concentrations tested for a given construct; the "WT" control construct has been used multiple times (n > 3). **e** Sequence of the 20-nucleotide region upstream of the 5′ss of *HTT* stop-codon psiExon49a showing partial deletions and mutations performed on this region, and their effects on HTT-C2 induced splicing (lower panel). Through partial deletion or mutation of the nucleotides CAGGA at positions −38 to −34, this region was shown to be important in regulating splicing events. The data are from a single transfection experiment with multiple concentrations tested for a given construct; the "WT" control construct has been used multiple times (n > 3).

equivalent mHTT protein lowering in both brain and peripheral tissues in two mouse models carrying human m*HTT* transgene, BACHD[20] and Hu97/18[24] (Fig. 5f). In BACHD mice, we also assessed HTT-D3 activity on mHTT splicing. BACHD mice were given either 1 dose (single dosing) or 21 consecutive daily doses (multiple dosing) of 10 mg/kg HTT-D3. We observed a similar response in mHTT mRNA and protein levels in BACHD brain after compound treatment (Supplementary Fig. 9d). In the Hu97/18 model, we observed uniform mHTT protein reduction in two critical brain sections, the striatum and cortex (Fig. 5f). Brain and cerebrospinal fluid (CSF) mHTT levels were significantly correlated (Fig. 5g). A similar correlation was observed between plasma and CSF mHTT levels upon HTT-D3 treatment (Fig. 5g). This underscores the advantage of advancing molecules with reduced efflux, such as HTT-D3, as potential HTT-lowering therapeutics for HD, where uniform HTT lowering is observed throughout the body.

## Discussion

HD is a devastating disorder with no approved treatment. Several preclinical studies support the hypothesis that targeting the expression of m*HTT* may prevent and/or slow disease progression[25–29]. To our knowledge, this work represents the first example of the identification and optimisation of orally bioavailable splicing modifiers that penetrate all tissues (including every cell type throughout the brain and periphery) and exert their action on m*HTT* splicing and protein lowering evenly throughout the body. A molecule from our HD drug discovery programme has recently entered a phase 1 clinical trial.

We initially discovered molecules such as HTT-C2 that lowered mHTT levels in the CNS. However, 50% lowering in the brain resulted in near depletion of mHTT (>90%) in peripheral tissues. Ablation of wt HTT in mice has been reported to cause acute pancreatitis due to degeneration of pancreatic acinar cells[22]. We undertook a chemical optimisation campaign that led to the

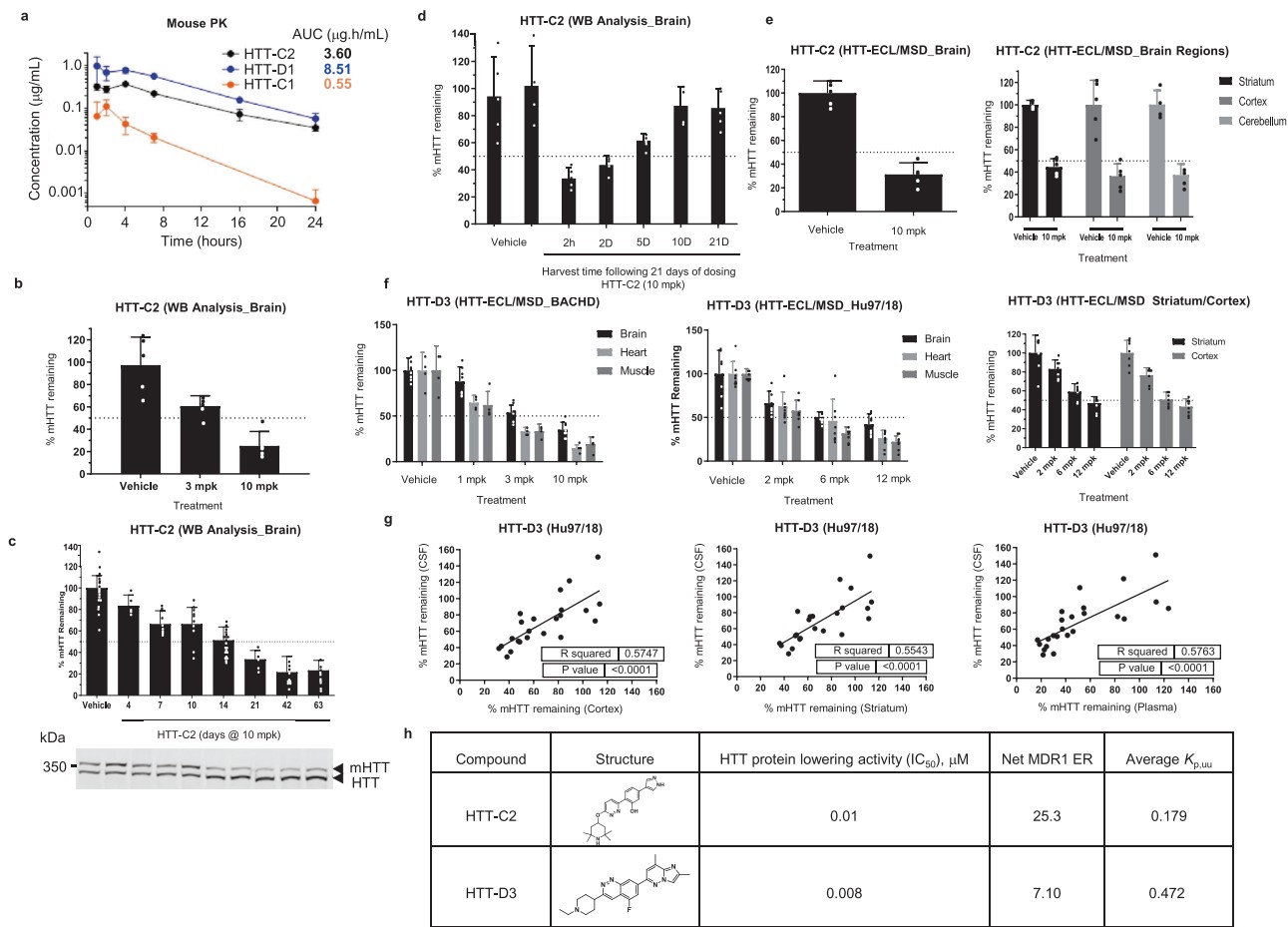

**Fig. 5 Compounds eliciting human huntingtin (HTT)-lowering in vivo. a** Plasma levels of HTT-C1, HTT-D1 and HTT-C2 over 24 h in BACHD mice after a single 10 mg/kg dose. **b** Western blot analysis of human HTT protein within the brain tissue of BACHD mice treated with HTT-C2 (3 mg/kg or 10 mg/kg) once daily for 14 days; graph shows percent lowering relative to vehicle control and normalised to mouse Htt protein. Data represent mean ± SD (error bars) of five animals per data point. **c** Western blot analysis of 10 mg/kg HTT-C2 induced lowering of human HTT protein within brains of BACHD mice over time. Graph shows percent lowering relative to vehicle control and normalised to mouse Htt protein. Example western blot shown below graph with mouse Htt as a loading control. Data represent mean ± SD (error bars) of five animals per data point. **d** Western blot analysis of human HTT protein expression levels in brain tissue over time following cessation of 10 mg/kg HTT-C2 treatment in BACHD mice. Graph shows percent lowering of human HTT protein relative to vehicle control and normalised to mouse Htt protein. Data represent mean ± SD (error bars) of five to eight animals per data point. **e** Electrochemiluminescence (ECL) analysis of human HTT protein expression levels within different parts of the brain from BACHD mice treated with 10 mg/kg HTT-C2. Graphs show percent lowering relative to vehicle control and normalised to utrophin (UTRN). Data represent mean ± SD (error bars) of five to eight animals per data point. **f** ECL analysis of human HTT protein expression levels within different tissues, including the striatum and cortex, of Hu97/18 mice and BACHD mice treated with HTT-D3. Graphs show percent lowering relative to vehicle control and normalised to Kirsten rat sarcoma viral oncogene homologue (KRAS). Data represent mean ± SD (error bars) of five to eight animals per data point. **g** Regression analysis to show the correlation between HTT lowering of HTT-D3 in the cortex, striatum and plasma of Hu97/18 mice relative to the cerebrospinal fluid (CSF). Correlations were analysed (using GraphPad Prism) by linear regression with $R^2$ and $P$ values indicated on graphs. $P$ values of <0.05 were considered statistically significant. **h** Chemical structures of HTT-C2 and HTT-D3.

identification of advanced molecules with improved pharmaceutical properties. For example, HTT-D3 has even distribution throughout the body and results in correlative and equal reduction of mHTT protein levels in plasma and CSF of Hu97/18 mice. This is a highly desirable profile for oral HD therapy.

The compounds described here lower wt and m*HTT* mRNA and protein levels by inducing the inclusion of a stop-codon psiExon with a noncanonical 5′ss into *HTT* mRNA, reducing mRNA levels through NMD and preventing full-length protein production (International Patent Application Publication Nos. WO 2017/100726 A1, WO 2018/098446 A1, WO 2018/232039 A1, WO 2019/005980 A1 and WO 2019/005993 A1; see also Ratni et al. 2018[9]). Recently, Ando et al.[30] described compounds acting via a similar mechanism.

The strong preference for the noncanonical AGAguaag 5′ss and the similarity of this motif to that recognised by risdiplam provide compelling evidence that HTT-C2 and analogues function as molecular glues to strengthen U1 small nuclear ribonucleoprotein (snRNP) and pre-mRNA 5′ss interaction, allowing exon recognition by the spliceosome. Furthermore, by utilising a U1 variant fully complementary to the GAgu 5′ss, we demonstrate that in addition to the set of compound-induced psiExons, the U1 variant promoted the inclusion of many additional psiExons throughout the genome. This result further demonstrates the stringency and specificity for adenosine at both the −3 and +3 positions at the 5′ss for compound-induced splicing. Therefore, at concentrations substantially higher than the HTT-lowering IC₅₀, these molecules do not cause widespread changes

in gene expression. Together with our results highlight the critical recognition step in pre-mRNA splicing and how it can be manipulated by small molecules. The requirement for a predicted ESE sequence within *HTT* stop-codon psiExon49a demonstrates the importance of exon definition in the activation of potential compound-induced psiExons. As has been shown for exonisation of Alu elements, mutation to strong 5′ and 3′ss is not sufficient for exon recognition by the spliceosome; additional factors are required, such as serine/arginine-rich protein-binding sites and ESEs established over thousands to millions of years[31]. The small number of psiExon inclusion events, identified by U1-GA variant and compound treatment demonstrates this requirement for exon definition and highlights the high specificity of our *HTT*-selective compounds, which activate only 31 of >58,000 potential psiExons.

Finally, through comparison of RNA-Seq from SMN-C3 to that of HTT-C2, we identify a strong preference for A at the −4 and −3 positions of the 5′ ss, respectively, defining two distinct classes of splicing modifiers that target the GAgu noncanonical 5′ ss. Since ~45% of known human exons contain a noncanonical 5′ ss in the −2, −1 position (non-AG; Supplementary Fig. 10), including GA (1.3% of total Refseq defined exons), our results suggest that additional noncanonical 5′ss could potentially be targeted by small-molecule splicing modifiers that remain to be discovered. The insights from our work can lead to the discovery of therapeutics for diseases of unmet medical need.

## Methods

**Cell culture**. Human B-lymphocytes and fibroblasts derived from the same homozygous patient with Huntington's disease (HD) (GM04856/GM04857) and a healthy donor (GM07492/GM07491) (Coriell Institute for Medical Research), human neuroblastoma (SH-SY5Y) cells (ATCC®), human embryonic kidney 293 (HEK293) cells (ATCC); Madin–Darby Canine Kidney (MDCK) cells (ATCC®); MDCK cells expressing multidrug-resistance mutation 1 (MDCK-MDR1) (Absorption Systems); mouse CT26 cells (ATCC) were all grown at 37 °C in a humidified 5% $CO_2$ atmosphere. Fibroblasts were maintained in Dulbecco's Modified Eagle's Medium (DMEM) with 10% (v/v) fetal bovine serum (FBS) (Thermo Fisher Scientific) and 1% antibiotic cocktail (penicillin-streptomycin/ Thermo Fisher Scientific). All cell lines tested negative for mycoplasma contamination. Following the purchase of the cell lines, none of the cell lines was further authenticated. No commonly misidentified cell lines were used in the study.

**High-throughput screening for the identification of HTT-lowering molecules**. Human fibroblasts derived from a homozygous patient with HD (GM04857) were grown for 96 h in the presence of test compounds (in 0.5% DMSO) or controls at 37 °C in 5% $CO_2$. After 96 h, cells were lysed and frozen. Huntingtin protein (HTT) levels were measured in lysates as described below. Compounds that decreased HTT levels relative to DMSO control were further tested in full dose–response.

**Quantification of HTT protein in cells**. For analysis in the ECL assays, test compounds were serially diluted with a threefold step in 100% DMSO (Sigma®) to generate a seven-point concentration–response curve. A solution of test compound (500 nL, 200× in DMSO) was added to each test well; the final concentration of DMSO was 0.5%. Fibroblasts were seeded in 96-well flat-bottomed plates (Thermo Fisher Scientific) at $4 \times 10^3$ cells/well in 100 μl of culture medium containing the test compound or DMSO vehicle control and incubated for 96 h (37 °C, 5% $CO_2$, 100% relative humidity). After removal of the supernatant, cells were lysed in 50 μL of 1× Lysis Buffer11 (LB11) extraction buffer (50 mM Tris [pH 7.4], 300 mM NaCl, 10% [w/v] glycerol, 3 mM ethylenediaminetetraacetic acid (EDTA), 1 mM MgCl₂, 20 mM glycerophosphate, 25 mM NaF, 1% Triton X-100), containing a Complete™ protease inhibitor cocktail (Roche Diagnostics®) with shaking at 4 °C for 30 min; the plates were then stored at −20 °C.

For western blot analysis, fibroblasts were plated at $5 \times 10^4$ cells/well in 1 mL 10% FBS/DMEM with GlutaMAX™ supplement (Thermo Fisher Scientific) in 24-well plates (Thermo Fisher Scientific) and incubated for 3–4 h (37 °C, 5% $CO_2$, 100% relative humidity). Cells were then treated with HTT-C1 at different concentrations (0.5% DMSO) in triplicate wells for 96 h. Cells were lysed in 75 μL Laemmli buffer (Bio-Rad Laboratories, Inc.).

**ECL protein assay**. MESO SCALE DISCOVERY® 96-well plates (MSD®) were coated overnight at 4 °C with primary antibodies in phosphate-buffered saline (PBS; 30 μl/well). The plates were washed three times with 0.05% Tween-20 in 1× PBS (PBS-T; 200 μl/well) then blocked (100 μl/well; 5% bovine serum albumin (BSA) in PBS-T) for 5–6 h at room temperature with shaking. Plates were then

washed three times with PBS-T. Cell lysates were transferred to the antibody-coated plates (25 μl/well) and incubated with shaking overnight at 4 °C. After removal of the lysates, the plates were washed three times with PBS-T, and 25 μl of detection antibody in 1% BSA, PBS-T was added to each well and incubated with shaking for several hours at room temperature. After three washes with PBS-T, 25 μl of SULFO-TAG secondary antibody (MSD; 0.25 μg/ml in 1% BSA, PBS-T) was added to each well and incubated with shaking for 1 h at room temperature. After washing three times with PBS-T, 150 μl of read buffer T with surfactant (MSD) was added to each empty well and the plate was imaged on the SI 6000 imager (MSD) according to manufacturers' instructions for 96-well plates. Primary capture antibodies included: anti-polyglutamine-expanded HTT mouse monoclonal antibody (mAb) at 1 μg/mL (clone MW1; Developmental Studies Hybridoma Bank); anti-HTT mAb at 1 μg/mL (clone 1HU-4C8; Millipore; catalogue # MAB2166); anti- Kirsten rat sarcoma viral oncogene homologue (KRAS) rabbit polyclonal antibody at 1 μg/ml (Abcam; catalogue # ab137739). Detection antibodies included: Huntingtin (D7F7) XP® Rabbit mAb at 0.25 μg/ml (Cell Signalling Technology®; catalogue # 5656); anti-human-KRAS mAb at 0.25 μg/ml (clone 2C1; LSBio; catalogue # LS-C175665-100).

**RT-qPCR quantification of HTT mRNA in cells**. Test compounds were serially diluted threefold in 100% DMSO to generate a seven-point concentration curve. A solution of test compound (500 nL, 200× in DMSO) was added to each test well. Fibroblasts were seeded in 96-well flat-bottomed plates (Thermo Fisher Scientific) at $1 \times 10^4$ cells/well in 100 μl of culture medium containing the test compound or DMSO vehicle control and incubated for 24 h (37 °C, 5% $CO_2$, 100% relative humidity). After removal of the supernatant, cells were lysed in RNA lysis buffer (1 M Tris-HCL pH 7.4, 5 M NaCl, 10% IGEPAL®CA-630; 50 μL/well) for 1 min at room temperature, before 50 μL of chilled nuclease-free water was added to each well; plates were then transferred immediately onto the ice before storing at −80 °C overnight.

The mRNA levels of the huntingtin gene (*HTT*) and glyceraldehyde-3-phosphate dehydrogenase (*GAPDH*) were quantified using TaqMan-based RT-qPCR primers and probes (Thermo Fisher Scientific; Supplementary Table 1) and the AgPath-ID™ one-step RT-PCR Kit (Thermo Fisher Scientific). RNA samples were transferred (2 μL/well) to the Armadillo 384-well PCR plate (Thermo Fisher Scientific) containing 8 μL/well of the AgPath-ID™ one-step RT-PCR reaction mixture (Thermo Fisher Scientific) in a final volume of 20 μL. The plate was then sealed with MicroAmp™ Optical Adhesive Film (Thermo Fisher Scientific) and placed in the CFX384 Touch™ Real-Time PCR thermocycler (Bio-Rad Laboratories, Inc.). RT-qPCR was carried out at the following temperatures for indicated times: Step 1: 48 °C (30 min); Step 2: 95 °C (10 min); Step 3: 95 °C (15 s); Step 4: 60 °C (1 min); then Steps 3 and 4 were repeated for a total of 40 cycles.

**Western blot analysis**. HD fibroblasts (GM04857) were treated with compounds and lysed in 70 μL sample buffer (24-well plates/Thermo Fisher Scientific). Medium was aspirated, rinsed once with PBS, 70 μL Invitrogen™ NuPAGE™ Sample Buffer was added, rocked at room temperature for 10 min, and lysates frozen. Cell lysates were boiled for 10 min—loaded 45 μL/well. Cell lysates (45 μl/lane) were separated using pre-cast 3–8% Tris-Acetate gels (NuPAGE™ 3–8%, Tris-Acetate, 1.0 mm, Midi Protein Gel, 12 + 2-well; Thermo Fisher Scientific) for 5–6 h at 130 V. After electrophoresis, proteins were transferred to nitrocellulose membranes (0.45 μM nitrocellulose/Bio-Rad) at 150 mAp in NuPAGE Transfer Buffer (20× diluted to 1×; Thermo Fisher Scientific) for 90 min at 4 °C. Membranes were blocked overnight in blocking buffer (Li-Cor blocking buffer + 0.1% Tween) at 4 °C, washed four times (for 5 min) in PBS + 0.1% Tween (PBS-T; Thermo Fisher Scientific), and incubated overnight at 4 °C with primary antibodies (in Li-Cor blocking buffer + 0.1% Tween). Blots were washed with PBS-T, probed with secondary antibodies (in Li-Cor blocking buffer + 0.1% Tween) at room temperature for 1 h and washed again with PBS-T. Bound antibodies were visualised using the Odyssey® imaging system (LI-COR) according to the manufacturer's instructions. The following primary antibodies were used: anti-HTT at 1:1000 (clone 1HU-4C8; Millipore; catalogue # MAB2166), anti-utrophin (UTRN) at 1:250 (clone DRP3/20C5; Vector Laboratories; catalogue # VP-U579), anti-oxidoreductase-protein disulphide isomerase (PDI) at 1:10,000 (Santa Cruz; catalogue # SC20132), anti-βactin at 1:10,000 (clone AC-74; Sigma; catalogue # A2228), anti-GAPDH at 1:1000 (Thermo Fisher Scientific; catalogue # PA1-987), anti-alpha serine/threonine-protein kinase (AKT) at 1:1000 (Cell Signalling; catalogue # 9272). Secondary antibodies included: Alexa Fluor® 680 goat anti-mouse immunoglobulin G (IgG) at 1:10,000 (Thermo Fisher Scientific; catalogue # A28183), IRDye® 800CW donkey anti-mouse IgG at 1:10,000 (Li-Cor; catalogue # 926-32212), and IRDye® 800CW donkey anti-rabbit IgG at 1:10,000 (Li-Cor; catalogue # 925-32213).

**Primer walking assay, endpoint RT-PCR and AmpliSeq analysis**. B-lymphocytes (GM04856 cells) were plated in six-well plates (Thermofisher) at $5 \times 10^5$ cells/well in 2 mL of 10% FBS, DMEM and incubated for 6 h (37 °C, 5% $CO_2$, 100% relative humidity). Cells were then treated with HTT-C1 at 125 nM (in 0.5% DMSO) in triplicate for 24 h. RNA was purified with the RNeasy Mini Kit (Qiagen) using the manufacturer's protocol. Samples were prepared for RT-PCR

(described previously in Methods) using 0.04 μL of each primer (at 100 μM). For reverse transcription and PCR, the following steps were performed: reverse transcription step: 48 °C (15 min); PCR steps: Step 1: 95 °C (10 min), Step 2: 95 °C (30 s), Step 3: 55 °C (30 s), Step 4: 68 °C (1 min); Steps 2–4 were repeated for 34 cycles, then held at 4 °C. PCR products were separated on 2% pre-cast agarose E-gels (Invitrogen), stained with ethidium bromide and visualised using a UVP gel imager (Thermo Fisher Scientific). Primer sets used for primer walking are provided in Supplementary Table 2.

Ion AmpliSeq analysis of *HTT* pre-mRNA splicing GM04856 cells were plated in six-well plates at $5 \times 10^5$ cells/well in 2 mL of 10% FBS, DMEM and incubated for 6 h (37 °C, 5% $CO_2$, 100% relative humidity) before treatment with 125 nM HTT-C1 (in 0.5% DMSO) in triplicate for 24 h. Cellular RNA was then extracted and purified (described previously in Methods). The Ion AmpliSeq technology (Thermo Fisher Scientific), which is a PCR-based target enrichment and next-generation sequencing platform, was used as a targeted measure of exons across the entire *HTT* transcript, with the goal of monitoring the presence of any (and all) novel splice isoforms in the presence of compound. PCR enrichment of *HTT* exon targets was accomplished by applying a custom *HTT* AmpliSeq panel (PTC proprietary primer set). The panel consisted of two separate PCR primer pools, each producing 33 amplicons. The complete *HTT* assay has 66 amplicons (mean size, 135 bp) covering all 67 exons. The AmpliSeq workflow (Supplementary Fig. 11) included: (a) RNA reverse transcription, (b) target amplification, (c) partial primer digestion, (d) adapter ligation, (e) library amplification, (f) sequencing reaction, (g) sequencing data analysis. For data analysis, AmpliSeq reads (Fastq format) were mapped to human genome (hg19) using TopHat2[32] which allows identification of both known and novel splice junctions. For each one of the 66 introns of the *HTT* gene, we calculated a JEI (Fig. 2c) using the percent of read supporting the splicing of the exact annotated intron among all reads supporting the splicing isoforms using either the 5′ss or/and 3′ss of that intron. A JEI value of 100% indicates full splicing of the intron. A JEI value <100% indicates alternative splicing paths (eg. inclusion of a cryptic exon or use of alternative 5′ or 3′ss). Biological triplicates were performed for each treatment group. We compared the difference of JEIs between compound and DMSO treated samples using the Student's *t* test.

**Splice-site score.** The 5′ and 3′ss MAXENT scores were calculated using MaxEntScan[33] (http://hollywood.mit.edu/burgelab/maxent/Xmaxentscan_scoreseq.html) representing the strength of splice sites.

**Analysis of nonsense-mediated decay.** GM04856 cells were treated with DMSO or 250 nM HTT-C1. After 18 h, cells were treated with 10 μM cycloheximide (Sigma) or DMSO. Total RNA was isolated after 2, 4 and 8 h and analysed by endpoint PCR (described previously in Methods).

**Transfection.** Wild-type and mHTT and U1 minigene constructs were designed at PTC and synthesised at GenScript®. For U1 constructs $5 \times 10^5$ HEK293 cells were transfected with 2 μg of plasmid deoxyribonucleic acid (DNA) or mock control in six-well plates, using 6 μl Fugene6® (Promega) according to the manufacturer's instructions; after incubating for 24 h (37 °C, 5% $CO_2$, 100% relative humidity), cells were treated with either 1 μM HTT-C1 or 0.5% DMSO control and incubated for 48 h. For *HTT* constructs, $5 \times 10^5$ HEK293 cells were transfected with 50 ng of plasmid DNA in 24-well plates, using 6 μl Fugene6® according to the manufacturer's instructions. After incubating overnight (37 °C, 5% $CO_2$, 100% relative humidity), cells were treated with varying concentrations of compounds in a final concentration 0.05% DMSO and incubated for 24 h.

**RNA-Seq library preparation from SHY5Y and U1 transfected HEK293.** SHY5Y cells were seeded in six-well plates at $6 \times 10^5$ cells/well in 2 mL 10% FBS, DMEM and incubated for 4 h. Cells were then treated with two biological replicates of HTT-C1 at 24 nM or 100 nM (in 0.1% DMSO), or four biological replicates of vehicle control (DMSO) for 24 h (37 °C, 5% $CO_2$, 100% relative humidity). HEK293 cells were transfected with U1-GA variant minigene construct or mock for 48 h (37 °C, 5% $CO_2$, 100% relative humidity).

Total RNA was extracted using the RNeasy Plus Mini Kit. RNA concentration and quality were assessed using a NanoDrop spectrophotometer (ThermoFisher Scientific). For library preparation and sequencing, mRNA was enriched from ~3 μg of total RNA using oligo(dT) beads. The mRNA was fragmented randomly using fragmentation buffer followed by complementary DNA (cDNA) synthesis using an mRNA template and random hexamers primer. Second-strand synthesis buffer (Illumina), deoxynucleotides, ribonuclease H and DNA polymerase I were added to initiate second-strand synthesis. After a series of terminal repair, A-ligation and sequencing adaptor ligation, the double-stranded cDNA library was completed through size selection and PCR enrichment. RNA libraries were sequenced in a HiSeq sequencer (Illumina).

**RNA-Seq analysis of pre-mRNA splicing.** RNA sequencing reads were mapped to human genome (hg19) using STAR (version 2.5)[34]; only uniquely mapped reads (with MAPQ > 10) with <5nt/100nt mismatches and properly paired reads were used. For gene expression analysis, the number of reads in the coding sequence

(CDS) region of protein-coding genes and exonic region of non-coding genes were counted and analysed using DESeq2[35] (Bioconductor). For splicing analysis, all junction reads (read with a gap in alignment indicating splicing) were used, including the ones mapped to unannotated splice sites. Reads were counted for different exons (for cassette exon [CE]) or exonic regions (for A5′ss or A3′ss). For each splicing event, a percent-spliced-in (PSI) value was calculated using the percent of average read number supporting the inclusion among all reads supporting either the inclusion or the exclusion. A minimum of 20 for the denominator of PSI calculation was required. Otherwise, a 'NA' value would be generated. PSI values for biological replicates were averaged, and the PSI difference between the two treatment groups was calculated. For a statistical test, a $2 \times 2$ read counts table was made for each event, with rows for reads supporting inclusion or exclusion, and columns for the two comparing sample groups (biological replicates were combined). Fisher's exact test was used for the statistical tests. PSI change of >20% (or < −20%) and $P < 0.001$ was used to select splicing events being regulated by the treatment. PSI calculation, Fisher's exact test, k-mer analysis and statistical analysis were performed using R (3.5.1).

**K-mer analysis.** For comparing sequence difference of a particular region for two groups of exons (e.g., Inc vs. NC) we compared the k-mer ($k = 4$–6) frequencies of the two groups by Fisher's Exact Test (one k-mer vs. all other k-mers, Group 1 vs. Group 2). The resulting $P$ value was converted to a significance score ([SS] $= -S*\log_{10} P$ value), in which S is the sign indicating enrichment (1) or depletion (−1) of the k-mer in Group 1.

**Sequence logo.** Sequence logos were generated using WebLogo[36] (University of California, Berkeley).

**Genome-wide identification of putative GA-psiExons.** We searched the human genome with "AGAgtaag" sequences (potential 5′ss sequence responding to the compound) within introns of ReSeq annotated genes. Then 3′ss sequences were scanned upstreaming of the "AGAgtaag" sequence using MaxEntScan[33]. A putative GA-psiExon is defined as an unannotated exon with length between 6 and 200nt and 3′ss score >2.3.

**Minigene constructs.** HEK293 cells transfected with minigene constructs were treated with varying concentrations of test compounds in a final concentration 0.05% DMSO and incubated for 24 h. Total RNA was isolated from the cells using the RNeasy Plus Mini Kit (Qiagen) and RNA concentration and quality were assessed using a NanoDrop spectrophotometer (Thermo Fisher Scientific). To determine splicing changes, cDNA was synthesised using the iScript™ cDNA synthesis kit (Bio-Rad Laboratories) according to the manufacturer's instructions. Endpoint PCRs were set up using Platinum™ PCR SuperMix High Fidelity (Invitrogen) and the resulting PCR products were separated on 2% E-gels (Invitrogen). Primers (designed by PTC Therapeutics, supplied by Invitrogen) were directed against common sequences in the minigene constructs: T7 Forward: 5′-TAATA CGACTCACTATAGGG-3′; BGH Reverse, 5′-TAGAAGGCACAGTCGAGG-3′.

**Animal studies.** All in-life animal procedures were performed in a laboratory certified by the American Association for the Accreditation of Laboratory Animal Care (AAALAC) with approval from the Institutional Care and Animal Use Committee. BACHD[20] and Hu97/18[24] mice were used, and the genotype of each BACHD animal was confirmed by an in-house PCR assay prior to enrolment in the study.

**Quantification of HTT protein in animal tissues.** Test mice were euthanised, and brain, muscle (quadriceps), other peripheral tissues and blood samples were harvested 2 h after the last dose on Day 20. Prior to analysis, crude total protein from brain and peripheral tissue samples were prepared by sample lysis in MSD assay buffer 1 (MSD) with Complete™ Protease Inhibitor Cocktail added (Roche Diagnostics). Tissues were then homogenised by TissueLyser II (Qiagen), plus a 5 mm stainless steel bead. The lysate was clarified by centrifugation at $16,000 \times g$ for 20 min at 4 °C, and the total protein concentration quantified with the Pierce™ BCA Protein Assay Kit (Thermo Scientific), according to the manufacturer's instructions. Whole blood was collected by cardiac puncture into EDTA collection tubes. An aliquot (100–200 μL) was added to 1.5 mL of eBioscience™ 1v Red Blood Cell Lysis Buffer (Thermo Fisher) and mixed well for 5 min before collecting the white blood cells (WBC) by centrifugation at $400 \times g$. The supernatant was discarded, and the pellet of WBCs was frozen in liquid nitrogen and stored at −70 °C. Brain and peripheral sample lysates were analysed for hHTT and KRAS protein expression using the ECL protein assay (described previously in Methods); using the same method WBC samples were also analysed for hHTT expression but not KRAS.

Each tissue sample was tested in duplicate using the ECL protein assay and the average hHTT and KRAS readouts were calculated. The ratio of the mean hHTT signal to the mean KRAS signal (×1000) was determined for each test animal. The hHTT/KRAS ratio grand mean for the vehicle group of five test animals was calculated, and the fold change relative to the vehicle grand mean was determined

for each test animal in each group. Percent hHTT (%hHTT) lowering normalised to KRAS was determined for each test animal by subtracting the fold change from one and multiplying the difference by 100. Each experiment was performed twice yielding ten %hHTT-lowering values for each treatment group. For each treatment group, the mean %hHTT lowering plus the standard error of the mean was plotted as a bar graph. The %hHTT lowering in WBC samples was determined without KRAS using the grand hHTT vehicle mean, instead of the grand hHTT/KRAS ratio vehicle mean.

**In vivo pharmacokinetic studies**. Oral pharmacokinetics (PK) of compounds were evaluated in wt littermates from the BACHD colony (FVB background). Mice were treated with test compounds (10 mg/kg) by oral gavage in 0.5% hydroxypropylmethyl cellulose (HPMC) with 0.1% Tween 80. Blood was collected by terminal cardiac puncture at specified time points (three mice per time point) and centrifuged to generate plasma. Brain tissue was collected at the time of blood collection and homogenised in water. Protein was precipitated from plasma and brain homogenates with acetonitrile, methanol mixture (5:1, v/v) containing an internal standard that is a close analogue of the test compounds. The mixture was filtered through an EMD Millipore MultiScreen™ Solvinert Filter Plate (MSRLN04, Millipore, Burlington, MA). Calibration standards were prepared in the same matrix and processed with the testing samples. Filtrates were analysed using an Acquity ultra performance liquid chromatography (UPLC) system (Waters Corporation) in tandem with Xevo TQ-s Spectrometer (Waters Corporation). Samples were injected on to a Waters UPLC Acquity BEH C18 Column (2.1 × 50 mm, 1.7 µm) maintained at 50 °C. The injection volume was 3 µL and the mobile phase flow rate was 0.45 mL/min. The mobile phase consisted of two solvents: a) 0.1% formic acid in water and b) 0.1% formic acid in acetonitrile. The initial mobile phase started with 5% solvent B for 0.4 min, which was changed to 98% solvent B over 0.8 min with linear gradients, and then maintained at 95% solvent B for another 0.4 min. The drug concentrations were acquired and processed with MassLynx 4.1 software. PK parameters were estimated using the non-compartment method within Phoenix® WinNonlin® Build 8.1 (Certara USA, Inc., Princeton, NJ).

**In vivo pharmacodynamic studies**. BACHD: pharmacodynamic (PD) evaluations were performed in BACHD mice aged 6−10 weeks. Compound or vehicle (HPMC/0.1% Tween 80) was administered to BACHD mice (five female mice per group) once daily for 21 doses (QD×21) by oral gavage; dosing volumes were 10 mL/kg. Each animal was regularly observed for mortality or signs of pain, distress or overt toxicity, and findings were recorded. Body weights were recorded at the start, and at least once a week, during the course of the study. Tissue samples were obtained and prepared for ECL protein assay analysis (described previously in Methods) from each animal.

Hu97/18: both sexes of 2–4-month-old Hu97/18 mice were used. Mice were maintained under a 12-hour light:12-hour dark cycle in a clean facility with free access to food and water. Experiments were performed with the approval of the Institute Animal Care and Use Committee of the University of Central Florida. Mice were treated with vehicle control or 2, 6 or 12 mg/kg of compound daily by oral gavage for 21 consecutive days. Mice were weighed 3x weekly and observed daily for general health and neurological signs, including gait, head tilt and circling. No adverse events were observed, and no mice were removed from the study.

**Hu97/18 terminal tissue and sample collection**. Mice were anaesthetised with Avertin (2,2,2-tribromoethanol, Sigma Aldrich, catalogue # T48402) and secured in a stereotaxic frame (Stoelting). The ear bars were raised and the nose piece used to position the mice in a manner that would allow for a near 90° tilt of the head to access the cisterna magna. A 1 cm² section of dorsal neck skin was removed, and muscle layers were completely dissected away to expose the cisterna magna, which was then cleaned with PBS and 70% ethanol and dried using compressed air. A 50cc Hamilton® syringe with point style 2 and a 12o bevel was then lowered carefully into the cisterna magna. CSF was slowly withdrawn at a rate of 10 µl/min using an UltraMicroPump with a Micro4 controller (World Precision Instruments). CSF samples were collected in pre-chilled tubes, centrifuged, then flash frozen in liquid nitrogen prior to storage at −80 °C.

Whole blood was then collected by cardiac puncture into EDTA-coated tubes and divided into three aliquots. One was immediately snap frozen, while plasma was isolated from another and crude peripheral blood mononuclear cells from the third. Mice were then decapitated, and the brain removed and placed in ice for ~1 min to increase tissue rigidity. During this interval, liver, heart and quadriceps muscle were isolated and snap frozen. Brains were then micro-dissected into cortex, hippocampus, striatum, cerebellum, and midbrain/brain stem.

**Immunoprecipitation and flow cytometry mtHTT quantification**. Approximately 10,000 5-µm carboxylate-modified latex beads (Invitrogen, catalogue # C37255) were coupled with capture antibody, HDB4E10 anti-HTT, in 50 µl of NP40 lysis buffer (150 mM NaCl, 50 mM Tris pH 7.4, Halt phosphatase (Thermo Scientific, catalogue # 78420) and Halt protease inhibitor cocktails (Thermo Scientific, catalogue # 78429), 2 mM sodium orthovanadate, 10 mM sodium fluoride NaF, 10 mM Iodoacetamide, and 1% NP40). Capture antibody coupled beads were then combined with 10 µl of CSF, or 20 µl of plasma in triplicate in a 96-well V-

bottom plate (Thermo Scientific, catalogue # 249944), brought to a total volume of 50 µl in NP40 lysis buffer, mixed well, and incubated overnight at 4 °C. The next day, the plate was spun down for 1 min at 650 RCF and supernatant was removed. Beads were washed three times in immunoprecipitation and flow cytometry (IP-FCM) wash buffer (100 mM NaCl, 50 mM Tris pH 7.4, 1% BSA, 0.01% sodium azide). MW1 anti-expanded polyglutamine probe antibody was biotinylated using EZ-Link Sulfo-NHS-Biotin (Thermo Scientific, catalogue # 21217), and 50 µl of the diluted antibody was incubated with the HDB4E10 beads bound to mtHTT for 2 h at 4 °C. Beads were washed three times with 200 µl of IP-FCM wash buffer. Streptavidin–phycoerythrin (PE) (BD Biosciences, catalogue # 554061) was prepared at 1:200 and 50 µl added to each well and incubated at RT, protected from light, for 30 min. Beads were washed three times with 200 µl of IP-FCM buffer, resuspended in 200 µl of IP-FCM wash buffer, and fluorescence intensity of ~2000 beads per sample, HDB4E10/MW1 mtHTT bead complexes, was measured using an Acuri C6 flow cytometer (BD Biosciences). Median fluorescent intensity of PE was measured for each sample to determine relative mtHTT protein levels.

**MDCK-MDR1 efflux assay**. The MDR1 efflux assay was conducted at Absorption System LLC (Exton, PA). In brief, MDCK-MDR1 and MDCK-wt cell monolayers were grown to confluence on collagen-coated, microporous membranes in 12-well assay plates (Thermofisher). Compound solutions (10 µM) in permeability assay buffer (Hanks' balanced salt solution [HBSS], 10 mM HEPES, 15 mM glucose; pH of 7.4) were placed in the donor chamber. The receiver chamber was filled with assay buffer plus 1% BSA. Cell monolayers were dosed on the apical side (A-to-B) or basolateral side (B-to-A) and incubated at 37 °C (5% CO₂, 100% relative humidity). Sampling from the donor chambers was performed at 0 and 1 hr; and from the receiver chambers at 1 hr. Each determination was performed in duplicate. The flux of lucifer yellow was also measured post-experimentally for each monolayer to ensure no damage was inflicted to the cell monolayers during the flux period. All samples were assayed by liquid chromatography-tandem mass spectrometry using electrospray ionisation. The apparent permeability ($P_{app}$) and percent recovery was determined using the following equation:

$$P_{app} = (\mathrm{d}C_r/\mathrm{d}t)^* V_r/(A \times C_0)$$

$\mathrm{d}C_r/\mathrm{d}t$ represents the slope of the cumulative receiver concentration vs. time in µM/s; $V_r$ is the volume of the receiver compartment (cm³); $V_d$ is the volume of the donor compartment in (cm³); A is the area of the insert (1.13 cm² for 12-well); $C_0$ is the average measured concentration of the donor chamber at time zero in µM; net efflux ratio is defined as $P_{app(B-to-A)} − P_{app(A-to-B)}$.

**Unbound brain partition coefficient ($K_{p,uu}$)**. The unbound brain partition coefficient ($K_{p,uu}$) is defined as the ratio between unbound brain-free drug concentration and unbound plasma concentration. It was calculated using the following equation:

$$K_{p,uu} = C_{brain} * f_{u,b}/(C_{plasma} * f_{u,p})$$

$C_{brain}$ and $C_{plasma}$ represent the compound concentrations in brain and plasma, respectively. $f_{u,b}$ and $f_{u,p}$ are the unbound fraction of each testing article in brain and plasma, respectively. Both $f_{u,b}$ and $f_{u,p}$ were determined in vitro using the Pierce device for rapid equilibrium dialysis at Absorption System LLC (Exton, PA).

$K_{p,uu}$ was calculated individually for each animal from multiple mouse PK studies and the average values are reported here.

**Quantification of human *HTT* mRNA in animal brain**. One group of transgenic BACHD mice in the study was orally administered a single dose of HTT-D3 at 10 mg/kg. One group (three mice) was administered vehicle alone on the same schedule. Dosing volumes were 10 mL/kg based on individual mouse weights. Dosing solutions were prepared once as the free base HTT-D3 dissolved in a vehicle comprising 0.5% HPMC and 0.1% TWEEN® 80 and stored at ambient temperature. Samples of brain tissues were obtained at 2, 4, and 8 h post HTT-D3 dosing. Samples were taken from the vehicle control mice at 2-h post vehicle dosing. Brain samples for each time point were obtained from three mice per group and total RNAs were prepared for analysis. Total RNAs from brain tissues were prepared following sample homogenisation and lysis in QIAzol Lysis Reagent using TissueLyser and the RNeasy Lipid Tissue Mini Kit (Qiagen #74804, Germantown, MD) according to the instructions provided in the manufacturer's kit.

Quantitative RT-qPCR was performed on ~50 ng of total RNA using AgPath-ID™ One-Step RT-qPCR Reagents (Life Technologies, Carlsbad, CA) according to the manufacturer's instructions using the custom TaqMan gene expression assays detailed in Supplementary Table 3.

An RT-qPCR reaction mixture of primers and probe sets for *HTT* and *GAPDH* was prepared according to Supplementary Table 4.

A volume of 2 µL of the RNA preps was transferred from each well to the Armadillo 384-well PCR plate containing 8 µL/well of the RT-qPCR reaction mixture that was prepared as detailed in Supplementary Table 4. The plate was then sealed with MicroAmp™ Optical Adhesive Film and placed in the C1000 thermocycler. The RT-qPCR was carried out at the following temperatures for the

indicated times: Step 1: 48 °C (30 min); Step 2: 95 °C (10 min); Step 3: 95 °C (15 s); Step 4: 60 °C (1 min); then, repeated Steps 3 and 4 for a total of 40 cycles.

The amplification efficiency was calculated from the slope of the PCR amplification curve for *HTT* and *GAPDH* individually. The abundances of *HTT* mRNA and *GAPDH* mRNA were then calculated as $(1 + E)^{-Ct}$, where Cycle threshold ($C_t$) is the threshold value for each amplicon. The abundance of *HTT* mRNA was normalised to *GAPDH* abundance. The normalised *HTT* mRNA level was then used to calculate the percent splicing in the HTT-D3-dosed group compared to the vehicle-treated group using a modified $2^{-\Delta\Delta CT}$ method[37].

**Reporting summary**. Further information on research design is available in the Nature Research Reporting Summary linked to this article.

## Data availability

The data supporting the findings of this study are available from the corresponding authors upon reasonable request. The AmpliSeq and RNA-Seq data generated in this study have been deposited in the GEO database under accession code GSE162814. The SMN-C3 RNA-Seq data was downloaded from the GEO database under accession code GSE62540. The code for the RNA-Seq data analysis can be found in GitHub (https://github.com/liwc01/DEDSeq). The following databases were also used: Refseq (https://www.ncbi.nlm.nih.gov/refseq/), Ensembl (https://useast.ensembl.org/index.html) and UCSC Known Genes (http://genome.ucsc.edu/). Source data are provided with this paper.

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

## Acknowledgements

Writing and editorial assistance was provided to the authors by Nicholas Black and Georgina Collett, PhD, on behalf of Lighthouse Medical Communications US LLC, New York, NY, USA, and funded by PTC Therapeutics, South Plainfield, NJ, USA. All authors met the ICMJE authorship criteria. Neither honoraria nor payments were made for authorship.

## Author contributions

A.B. conceived and designed the study. A.B., M.J., K.W., T.T., A.D and V.G. developed the assays. A.B., M.J. and T.T. performed the high-throughput screening. A.B., N.S., Y.C.M., G.K. and M.Wo analysed the hit selection/structure-activity relationship/chemistry. A.B., M.J. and K.W. performed the in vitro HTT protein and mRNA quantification and western blot analysis. A.B. and M.J. performed the primer walking assay. A.B., M.J., K.W., C.T., W.L., K.E. and J.G. performed the targeted next-generation sequencing. J.N., N.R., A.B., K.W., S.Y., Y.C., A.S. and M.H. performed the in vivo animal studies, drug metabolism and PK. W.L. and C.T. performed the bioinformatic analysis. W.L., C.T., A.B., K.E., N.N. and A.D. performed the statistical analyses and data interpretation. A.B., C.T., J.N., K.E., M.Wo, Y.C., W.L. and M.We drafted the manuscript. J.C., N.N., M.We, Y.C.M. and G.K. provided additional project support. S.P. provided full project support.

## Competing interests

Anuradha Bhattacharyya, Amal Dakka, Kerstin A. Effenberger, Vijayalakshmi Gabbeta, Minakshi B. Jani, Wencheng Li, Nikolai Naryshkin, Christopher R. Trotta and Kari J. Wiedinger are inventors in International Application Number PCT/US2018/037412, assigned to PTC Therapeutics, Inc., entitled 'Methods for Modifying RNA Splicing', relating to the use of HTT-C1. Anuradha Bhattacharyya, Minakshi B. Jani, Young-Choon Moon and Nadiya Sydorenko are inventors in U.S Patent 10,874,672, assigned to PTC

Therapeutics, Inc., entitled 'Methods for Treating Huntington's Disease', relating to the use of HTT-C1. Anuradha Bhattacharyya, Minakshi B. Jani, Nadiya Sydorenko and Matthew G. Woll are inventors in International Application Number PCT/US2018/039775, assigned to PTC Therapeutics, Inc., entitled 'Methods for Treating Huntington's Disease', relating to the use of HTT-D1. Nikolai Naryshkin is an inventor in U.S. Patent 10,195,202, assigned to PTC Therapeutics, Inc., entitled 'Methods for Modulating the Amount of RNA Transcripts', relating to the use of HTT-D1. Anuradha Bhattacharyya, Christopher R. Trotta, Jana Narasimhan, Wencheng Li, Kerstin A. Effenberger, Matthew G. Woll, Minakshi Jani, Nicole Risher, Shirley Yeh, Yaofeng Cheng, Nadiya Sydorenko, Young-Choon Moon, Gary M. Karp, Marla Weetall, Amal Dakka, Vijayalakshmi Gabbeta, Jason D. Graci, Thomas Tripodi, Jr., Joseph M. Colacino and Stuart W. Peltz are present employees of PTC Therapeutics, Inc., a biotechnology company. In connection with such employment, the authors received salary, benefits and stock-based compensation, including stock options, restricted stock, other stock-related grants and the right to purchase discounted stock through PTC's employee stock purchase plan. Kari Wiedinger and Nikolai Naryshkin are former employees of PTC Therapeutics, Inc. and hold stock in the company. Amber Southwell and Michael Hayden declare no competing interests.
