## [Peer Review File · Nature Communications]

Title: Small molecule splicing modifiers with systemic HTT-lowering activityREVIEWER COMMENTS

Reviewer #1 (Remarks to the Author):

The authors present the identification of novel small molecule splicing modifiers that lower HTT expression by promoting the inclusion of a pseudoexon containing a premature stop codon. They screened ~300,000 small molecules based on their ability to lower mutant HTT protein in HD fibroblasts and identified the splicing modifiers HTT-C1 and HTT-D1. They demonstrated that both molecules significantly reduce HTT mRNA and protein in HD fibroblasts in a dose-dependent manner.

To evaluate the transcriptome-wide effect of these compounds, they then performed RNA-sequencing and identified changes in gene expression and splicing after HTT-C2 treatment. They identified 31 new inclusion splicing events promoted by the treatment with no annotated splice site and enriched for the AGAguag sequence at the 5' ss. Inclusion of these exons leads to the generation of a stop codon with correlated downregulation of the host genes. Minigene splicing assays were used to confirm the importance of the AGAguag sequence to promote treatment response.

In vivo testing in mouse models carrying the human mutant HTT transgene (BACHD and Hu97/Hu18 mice) of HTT-D3, compound chemically optimized to achieve similar lowering of HTT between the periphery and the brain, demonstrated HTT protein reduction in two critical brain regions, striatum and cortex.

This manuscript is well-written and describes the identification of a promising therapeutic modality for HD, and it will be of interest to the field. It also defines a novel splicing-targeted mechanism that can be used to control gene expression. The clinical impact of lowering both normal and mutant HTT in patients remains unknown, however, this work represents a novel approach towards a much-needed therapy.

Major comments:

- 1) The authors should better explain the rationale behind the transitions between different compounds. For example, in the first Results section they describe the identification of two splicing modifiers, HTT-C1 and HTT-D1. They then went on to evaluate the global transcriptome changes of another compound HTT-C2, an HTT-C1 analog, without explaining why they used this specific compound rather than the two initially identified molecules. Later in the manuscript they mentioned that HTT-C2 has superior exposure in vivo compare with HTT-C1, however it should be clearer from the beginning the rationale behind compound selection for each experiment.
- 2) In the Figures and through the manuscript the authors should clearly specify if they are referring to mutant or WT HTT.
- 3) The RNA-seq analysis in HEK293 transfected with the U1-GA variant reprogrammed to bind non-canonical GAGu 5' ss only proves that the treatment acts by promoting the inclusion of weakly define

exons without demonstrating any HTT-C2 function on stabilizing the U1 interaction. In order to prove this, more specific MOA experiments would need to be performed (e.g. in vitro splicing assays comparing treated vs untreated HeLa nuclear extract). The authors should clearly explain that this is a potential activity, but more studies would be needed to confirm that the compound stabilizes U1.

4) In Figure 3d, there is a double band in the pseudoexon inclusion isoform lane in the panel representing the WT human HTT minigene. Has a cryptic splice site been activated? Every amplicon should be sequenced to verify their identity. The same is true for the panel below regarding the mutant CAGG_n minigene.

Minor comments:

1) The chemical structures of both HTT-C2 and HTT-D3 should be included in the main figures to make easier the comparison with the initial hit-compounds HTT-C1 and HTT-D1.

2) Please include the references for the sentence: "Several preclinical studies support the hypothesis that targeting the expression of HTT may prevent and/or slow disease progression." (rows 191-192).

Reviewer #2 (Remarks to the Author):

Huntingtin lowering methods in recent Huntington's disease (HD) clinical trials require invasive dosing regimens (e.g. intrathecal ASO injections every two months in the Roche/Ionis Phase III trial). Since Huntingtin lowering will likely be needed throughout life and beginning prior to the onset of neuronal degeneration, the development of non-invasive treatment options is urgently needed to increase the benefit/risk ratio. Bhattacharyya et al. describe the identification and characterization of small molecule splicing modulator compounds that reduce, in a dose-dependent manner, both normal and mutant HTT expression in cell and mouse models for Huntington's Disease. Importantly, the orally bioavailable compounds lower HTT levels throughout the CNS and the periphery, and act by promoting the inclusion of a pseudo-exon containing a translational termination codon (Stop-Codon psiExon) within HTT intron 49, leading to nonsense-mediated decay of the HTT mRNA and a reduction in the levels of HTT protein.

This is a timely study, considering the recent halting of both the Roche/Ionis and Wave Biosciences ASO clinical trials due to a potentially unfavorable benefit/risk ratio and lack of efficacy, respectively. Both trials require invasive intrathecal injections, and involved subjects that were clinically diagnosed with Huntington's disease. Due to the accumulating evidence that early and chronic therapeutic intervention will likely be needed in Huntington's disease, less invasive dosing options will now permit the design of trials involving pre-symptomatic treatment options. The findings of this study will be of interest, not only to those in the HD field, but also those involved in understanding the mechanism of splice site selection, alternative splicing, and the potential for splicing modulators to not only rescue gene expression (e.g. in SMA) but also reduce the expression of toxic genes in neurological disorders. The authors provide a

comprehensive analysis of the mechanism of action of their compounds (stabilize U1 snRNP interaction with specific sequences within the HTT intron 49), and statistical analyses were used appropriately in all experiments. Sufficient detail is also provided in their methods to permit reproduction of their work in other laboratories. Although there is always a concern for off-target effects of the splicing modulators that may be toxic, especially when administered chronically, the RNASeq data from SH-SY5Y cells is encouraging and shows that a relatively modest number of genes (165 and 215), including HTT, are affected by the HTT-C2 compound administered at 2X and 8X the effective inhibitory concentration.

Although determining the efficacy of the splicing modulators in vivo is beyond the scope of this study, it would be of interest to the HD field (and also strengthen the manuscript) to know if these compounds affect the level of mHTT exon 1 mRNA species that has been suggested to contribute to HD pathogenesis (Sathasivam et al., 2013). This truncated mRNA is hypothesized to be produced by impaired recognition of the mHTT intron 1 5' splice site by U1 snRNP (via sequestration of a U1 snRNP accessory protein by the CAG repeat) leading to premature polyadenylation within the intron. Translation of this short mRNA can produce an especially toxic mHTT exon 1-encoded fragment containing the expanded polyQ stretch.

Minor comments/suggestions:

1. The authors developed compound HTT-D3 to achieve similar lowering levels in the periphery and in the brain by selecting for efficient HTT lowering together with a reduced P-gp efflux (lines 168-189). The writing of this section is clear, but readers may benefit from the addition of more context by describing the importance of P-gp efflux in designing drugs targeting the brain (e.g. explain that P-gp function is responsible for MDR in cancer, and that expression P-gp is high not only in the periphery, but also in brain capillary endothelial cells).
2. The manuscript contains a large amount of data (and the authors have appropriately placed much of this data in supplementary figures). However, brevity required some very dense primary figures. Figure 1g,h show the plot of JEI following administration of vehicle or compound in several HTT introns, together with a schematic detailing the inclusion of the pseudo exon within intron 49 following treatment with the HTT-C1 compound. It would benefit the reader to separate these panels into a separate figure, and also include the schematic from Extended Data Fig. 9b.
3. Key data showing that the inclusion of the Stop Codon psiExon in intron 49 results in nonsense-mediated HTT mRNA decay (cycloheximide treatment and EP-PCR) is currently located in Extended Data Fig. 3. This data is important, and the authors should consider including it within a primary figure—for example, the new figure with the JEI plot. In the description of the cycloheximide experiment, the authors mention that total RNA was analyzed by error-prone PCR (EP-PCR) (described previously). A search of the manuscript did not reveal where this method was described previously, and a reference citation is needed.

Reviewer #3 (Remarks to the Author):

Review for “Small molecule splicing modifiers with systemic HTT-lowering activity”

The authors report interesting findings about identification, characterization, and in vivo pharmacodynamics of novel splicing modulator compounds that drive downregulation of HTT mRNA and protein levels. From their novel high-throughput screening campaign, they identify two molecules, HTT-C1 and HTT-D1, that induced decreases in HTT protein levels in patient fibroblast cell lines. They further characterize these changes as driven by alterations to HTT mRNA splicing – notably the inclusion of a poison (pseudo)exon between Exons 49-50 of the HTT mRNA sequence. This results in the triggering of nonsense-mediated decay of the mRNA and decreased protein expression, due to inefficient translation of the transcript. Evaluation of the mechanism of action demonstrates that the splicing modulation is comparatively selective, resulting in altered splicing of several hundred transcripts (similar to the effects seen with SMN splicing modulators). The authors then demonstrate in vivo pharmacodynamic changes in HTT protein in the brains of transgenic mouse models of Huntington’s disease (HD). The work concludes with the indication that a clinical trial in HD patients is in the process of initiating, promising the potential for benefits to patients with a severe neurodegenerative disorder.

Overall, this work provides an encouraging portrait of the expanding potential for both targeting of mRNA splicing for therapeutic benefit as well as the prospect for treatment of a devastating disease with no currently approved therapeutic options. This work represents a strong leap forward in the field and should be considered for publication, following the authors addressing the issues listed below.

Major Points:

- 1) The authors introduce compounds HTT-C1 and HTT-D1 in figure 1, and note their discovery in a HTT protein high-throughput screen. HTT-C1 and D1, are highly structurally similar to branaplam and risdiplam, respectively. As such, the description of the HTT alternative splicing/exon inclusion would be a most likely MoA of these molecules. In addition, there are several features of the SMN2 and HTT splicing that are comparable (e.g., both molecules induce inclusion of an exon that is normally poorly spliced and the GA at the -2 and -1 positions of the 5’ss). There is an extensive amount of the manuscript that is dedicated to describing the fundamental MoA of these compounds, when the work could benefit from a comparative analysis of the selectivity (rather than a repeat of the published work from the branaplam and risdiplam manuscripts). As such, the manuscript would benefit from a comparison/contrast between the respective MoA of these two molecule classes, and their respective target mRNAs, relative to the selectivity of the target gene/splicing.
 - a. To that end, an evaluation of the SMN2 splicing changes induced by HTT-C1 and HTT-D1 – and, conversely, evaluating risdiplam-mediated splicing of the poison pseudoexon in HTT would be important to demonstrate.
 - b. Additionally, the authors report on several alternative splicing events induced by HTT-C1. There is no data shown for the alternative splicing induced by HTT-D1, this would be important to demonstrate a similar phenotype, especially as the molecules are from different chemotypes. This seq data could also be compared/contrasted with the results from the prior publications on risdiplam and branaplam,

respectively to further bolster the selectivity of these molecules.

2) The authors show that – by RNAseq – there is an ~25% use of the poison pseudoexon junction reads, but show an ~80% decrease in protein level. How do they explain the discrepancy between a relatively minor depletion in the overall coding transcript for HTT (~75% full-length HTT – with no poison exon) and such a profound decrease in protein levels?

3) Much of the work focuses on HTT protein (and mRNA) lowering, but there is little discussion of the mutant vs. wild-type isoforms. As the poison exon is quite downstream of the mutant allele site (in exon 1), the compounds would be expected to impact both alleles comparably.

a. Use of a mutant-specific HTT antibody (e.g. MW1), for example, in the HTT Hu97/Hu18 would enable detection of the lowering of the mutant allele protein levels. As mutant HTT is known to form comparatively stable higher-order protein aggregates, it would be important to assess the relative pharmacodynamics (PD) of mutant HTT lowering. This information would be relevant in the context of assessment pharmacokinetic (PK)/PD relationships.

b. Further relevant to the PK/PD relationship would be a comparison of the relative mRNA and protein changes over time in vivo. The authors present data for in vivo dosing, and show that protein level drop does not occur until several days post-treatment – reaching a nadir at 21 days. They show some recovery of protein after 5 days, restoring to baseline by 10 days post-removal of treatment. No data is shown for mRNA changes over time and this would be important to demonstrate the “target-proximal” PD changes, relative to PK.

4) The Hu97/Hu18 mice have been shown to exhibit both pathologic and phenotypic alterations, due to the presence of the mutant HTT protein. This may be beyond the scope of the current manuscript, but the authors should consider presenting data from the mice as to the efficacy and tolerability of long-term treatment with HTT-D3. Treatment would be expected to reverse some of the phenotypes seen in these mice and evaluation of timing of treatment (e.g., juvenile vs. adult), relative to prevention vs. restoration of “wild-type” phenotypes (and their subsequent reversion to the “mutant” phenotypes) would be important.

Minor Points:

1) The authors note that, in mice with systemic depletion of HTT, there is evidence for pancreatitis. In the manuscript, they note that they switched the in vivo treatment paradigm from the HTT-C3 molecule to the HTT-D1 molecule, due to better partitioning between CNS and peripheral tissues, after an initial series of PK/PD experiments. Did the mice experience any pancreatic alterations due to a relatively transient lowering of HTT protein? This could be tracked by evaluation of amylase levels in serum, for example, and would be relevant for understanding of the relative onset of the phenotypes of HTT loss in CNS vs. peripheral tissues.

2) The authors note that the phenotype for these molecules is distinct from the SMN molecules (e.g., around line 212 in the discussion). Actually, the mechanism of action and phenotype is identical – both induce the inclusion of a poorly spliced exon. The difference is in the consequence of that exon inclusion. This is quite similar to the impact of alternative splicing for many splicing factors (e.g., HNRNP proteins) – as levels of these proteins rise, they induce alternative splicing of their own transcripts to trigger inclusion of poison exons to autoregulate the levels of their mRNA and protein. Some rephrasing around this section would help clarify this.

3) There are several minor questions around the alternative splicing:

a. Where the authors modified the sequence around the pseudoexon and the predicted ESE sequences, I wonder if their deletions and/or mutations altered any potential secondary structure that the RNA could form (e.g., stem loop) that could impede the access to U1.

b. Some comment could be made around the nature and consequences of the additional “off-mechanism” alternative splicing events seen in the RNAseq assessment. There was a summary statement about the number of exons and some comment about the presence of annotated and non-annotated exon events, but no further information was reported. A summary table with gene names, exon events, and predicted consequences for transcript & protein (e.g., does the new exon trigger frameshift & NMD, etc.) would be helpful.

Finally, it is important to note that – although it has not been reported in peer-reviewed literature, Novartis has indicated that they are repositioning branaplam in a Phase 2a study in HD patients (<https://huntingtonsdiseaseneews.com/2020/10/27/fda-gives-novartis-branaplam-orphan-drug-designation-for-huntingtons/>). From this, it is clear that the consideration of splicing modulation for treatment of HD is an idea that has gained acceptance.

Anuradha Bhattacharyya, PhD

PTC Therapeutics, Inc.

100 Corporate Court, South Plainfield, NJ, 07080

abhattacharyya@ptcbio.com

Chief Life Sciences Editor

14 May 2021

Dear editor, reviewers

We thank the reviewers for the careful and insightful review of our manuscript. Further to our resubmission today, please note that we are currently in the process of gathering available original data files/gels, and we anticipate having these available to send to you next week as a follow-up to our resubmission.

We have addressed all of the reviewer comments below – our response is indicated in blue.

Reviewer #1 (Remarks to the Author):

The authors present the identification of novel small molecule splicing modifiers that lower HTT expression by promoting the inclusion of a pseudoexon containing a premature stop codon.

They screened ~300,000 small molecules based on their ability to lower mutant HTT protein in HD fibroblasts and identified the splicing modifiers HTT-C1 and HTT-D1. They demonstrated that both molecules significantly reduce HTT mRNA and protein in HD fibroblasts in a dose-dependent manner.

To evaluate the transcriptome-wide effect of these compounds, they then performed RNA-sequencing and identified changes in gene expression and splicing after HTT-C2 treatment. They identified 31 new inclusion splicing events promoted by the treatment with no annotated splice site and enriched for the AGAguag sequence at the 5' ss. Inclusion of these exons leads to the generation of a stop codon with correlated downregulation of the host genes. Minigene splicing assays were used to confirm the importance of the AGAguag sequence to promote treatment response.

In vivo testing in mouse models carrying the human mutant HTT transgene (BACHD and Hu97/Hu18 mice) of HTT-D3, compound chemically optimized to achieve similar lowering of HTT between the periphery and the brain, demonstrated HTT protein reduction in two critical brain regions, striatum and cortex.

This manuscript is well-written and describes the identification of a promising therapeutic modality for HD, and it will be of interest to the field. It also defines a novel splicing-targeted mechanism that can be used to control gene expression. The clinical impact of lowering both normal and mutant HTT in patients remains unknown, however, this work represents a novel approach towards a much-needed therapy.

RESPONSE: We appreciate that reviewer 1 believes that this is a well-written manuscript and that it defines a novel splicing-targeted mechanism to modulate gene expression with oral small molecules. We thank the reviewer for their valuable comments.

Major comments:

1) The authors should better explain the rationale behind the transitions between different compounds. For example, in the first Results section they describe the identification of two splicing modifiers, HTT-C1 and HTT-D1. They then went on to evaluate the global transcriptome changes of another compound HTT-C2, an HTT-C1 analog, without explaining why they used this specific compound rather than the two initially identified molecules. Later in the manuscript they mentioned that HTT-C2 has superior exposure in vivo compare with HTT-C1, however it should be clearer from the beginning the rationale behind compound selection for each experiment.

RESPONSE: We appreciate this comment and agree that it is important to clarify from the beginning the rationale behind compound selection for each experiment.

Following hit confirmation, close analogs were evaluated for HTT lowering from both classes, one of which was HTT-C2 (branaplam). It was more potent and had superior exposure. Therefore, we decided to evaluate HTT-C2 in animals as well as analyze it for transcriptome-wide mRNA splicing or expression changes by RNA-sequencing. In addition, for these two classes of compounds, the inclusion of psiExons were not reported in the earlier publications.

Based on the reviewer's comment, we edited the following sentence in the section 'Global effects of splicing modification' in 'Results' (lines 105) of the revised manuscript.

Added the highlighted part in this sentence:

RNA-Seq analysis in human SH-SY5Y cells treated with HTT-C2 (a more potent, but structurally similar analogue of HTT-C1), or control (dimethyl sulfoxide [DMSO]) (Supplementary fig. 4a).

In addition, we have also added a section (lines 127–138) that profiles HTT-D1 like compounds like SMN-C3 (a closely related analog of risdiplam). In this section, we have reviewed our previously published RNA-Seq data (Naryshkin et. al). The revised analysis of the RNA-Seq data from SMN-C3 in the manuscript includes all the activated psiExons from this class.

2) In the Figures and through the manuscript the authors should clearly specify if they are referring to mutant or WT HTT.

RESPONSE: We appreciate this suggestion and have clarified throughout the revised manuscript when referring to mutant or wild type HTT in regards to a specific experiment/method.

3) The RNA-seq analysis in HEK293 transfected with the U1-GA variant reprogrammed to bind non-canonical GAgU 5' ss only proves that the treatment acts by promoting the inclusion of weakly define exons without demonstrating any HTT-C2 function on stabilizing the U1 interaction. In order to prove this, more specific MOA experiments would need to be performed (e.g. in vitro splicing assays comparing treated vs untreated HeLa nuclear extract). The authors should clearly explain that this is a potential activity, but more studies would be needed to confirm that the compound stabilizes U1.

RESPONSE: We appreciate this comment and agree that the objective of the U1 variant transient transfection experiment should be clearly stated.

We have rewritten this section in the section 'Global effects of splicing modification' in 'Results' (lines 139–153) of the revised manuscript to more clearly describe the U1 variant experiment and results. In addition, we have clarified the selectivity of compound effect vs U1 reprogramming.

4) In Figure 3d, there is a double band in the pseudoexon inclusion isoform lane in the panel representing the WT human HTT minigene. Has a cryptic splice site been activated? Every amplicon should be sequenced to verify their identity. The same is true for the panel below regarding the mutant CAGGgua minigene.

RESPONSE: We agree with the reviewer and have clarified below the identity of the bands in the figure.

In panel 4d (using updated numbering; formerly 3d), the wt human *HTT* minigene splicing induced by the compound does lead to two bands. The primary psiExon49a inclusion band and a minor larger inclusion. Based on size, this band does indeed represent a cryptic splice event, which is comprised of the use of the psiExon49a 5' splice site, but an alternative 3' splice site 31 nucleotides upstream of the predominant psiExon49a 3' splice site. Use of this alternative 3' splice site has been established by Ampliseq sequencing of the compound treated samples (please refer to the Sashimi plot in the new Fig. 2e. We also clarified this in the figure legend of Supplementary Figure 3).

For the GG construct, this alteration occurred in the absence of compound and the sizes correlate exactly with the events we see with the WT *HTT* minigene. We have included below the raw gel data for 3 constructs where it is clear when both WT and GG *HTT* minigene constructs are compared. Thus, these bands represent the splicing of psiExon49 to both versions of psiExon49a, just to a much large extent in the GG modification. This is expected as G at the -1 position is considered canonical and will interact with wild type U1 much more strongly due to increased base pairing. The additional upper band present in the GG modification may be the splice product of only splicing of the upstream intron of psiExon49a but the downstream intron (between psiExon49a and exon 50) is not spliced as indicated by the RNAseq data in SHSY5Y cells (see below).

Minor comments:

1) The chemical structures of both HTT-C2 and HTT-D3 should be included in the main figures to make easier the comparison with the initial hit-compounds HTT-C1 and HTT-D1.

RESPONSE: We agree with the reviewer and have added the structures of both HTT-C2 and HTT-D3 in the main figures section in the revised manuscript (figure 5i).

2) Please include the references for the sentence: “Several preclinical studies support the hypothesis that targeting the expression of HTT may prevent and/or slow disease progression.” (rows 191-192).

RESPONSE: We apologize for this and have added proper references to support that statement in the revised manuscript in the ‘Discussion’ (line 215–216).

Reviewer #2 (Remarks to the Author):

Huntingtin lowering methods in recent Huntington’s disease (HD) clinical trials require invasive dosing regimens (e.g. intrathecal ASO injections every two months in the Roche/Ionis Phase III trial). Since Huntingtin lowering will likely be needed throughout life and beginning prior to the onset of neuronal degeneration, the development of non-invasive treatment options is urgently needed to increase the benefit/risk ratio. Bhattacharyya et al. describe the identification and characterization of small molecule splicing modulator compounds that reduce, in a dose-dependent manner, both normal and mutant HTT expression in cell and mouse models for Huntington’s Disease. Importantly, the orally bioavailable compounds lower HTT levels throughout the CNS and the periphery, and act by promoting the inclusion of a pseudo-exon containing a translational termination codon (Stop-Codon psiExon) within HTT intron 49, leading to nonsense-mediated decay of the HTT mRNA and a reduction in the levels of HTT protein.

This is a timely study, considering the recent halting of both the Roche/Ionis and Wave Biosciences ASO clinical trials due to a potentially unfavorable benefit/risk ratio and lack of efficacy, respectively. Both trials require invasive intrathecal injections, and involved subjects that were clinically diagnosed with Huntington's disease. Due to the accumulating evidence that early and chronic therapeutic intervention will likely be needed in Huntington's disease, less invasive dosing options will now permit the design of trials involving pre-symptomatic treatment options. The findings of this study will be of interest, not only to those in the HD field, but also those involved in understanding the mechanism of splice site selection, alternative splicing, and the potential for splicing modulators to not only rescue gene expression (e.g. in SMA) but also reduce the expression of toxic genes in neurological disorders. The authors provide a comprehensive analysis of the mechanism of action of their compounds (stabilize U1 snRNP interaction with specific sequences within the HTT intron 49), and statistical analyses were used appropriately in all experiments. Sufficient detail is also provided in their methods to permit reproduction of their work in other laboratories. Although there is always a concern for off-target effects of the splicing modulators that may be toxic, especially when administered chronically, the RNASeq data from SH-SY5Y cells is encouraging and shows that a relatively modest number of genes (165 and 215), including HTT, are affected by the HTT-C2 compound administered at 2X and 8X the effective inhibitory concentration.

Although determining the efficacy of the splicing modulators in vivo is beyond the scope of this study, it would be of interest to the HD field (and also strengthen the manuscript) to know if these compounds affect the level of mHTT exon 1 mRNA species that has been suggested to contribute to HD pathogenesis (Sathasivam et al., 2013). This truncated mRNA is hypothesized to be produced by impaired recognition of the mHTT intron 1 5' splice site by U1 snRNP (via sequestration of a U1 snRNP accessory protein by the CAG repeat) leading to premature polyadenylation within the intron. Translation of this short mRNA can produce an especially toxic mHTT exon 1-encoded fragment containing the expanded polyQ stretch.

RESPONSE: We appreciate reviewer 2's comments and believe an HTT-lowering oral small molecule that is distributed uniformly throughout the brain and peripheral tissues is an attractive treatment option for HD. As the reviewer pointed out, this is a timely study and will be of major significance in the HD field. We thank the reviewer for their valuable comments.

The reviewer raised a very interesting point regarding the effect of our splicing modifiers on the level of the aberrantly spliced mHTT exon 1 mRNA species. In the figure below, we present RNA-Seq data (unpublished) using HD patient lymphocyte and cortical and striatal tissues from BACHD mice (the full-length-HTT HD mouse model). The data shows no obvious mHTT intron 1 polyadenylation transcript expressed in these tissues (please examine the reads between exon

1 [E1] and 2 [E2]) in these two model systems. Thus, we are not able to evaluate if these compounds affect the level of mHTT exon 1 mRNA species in our model systems. The models described in earlier publications (R6/2 and zQ175) do not have human *HTT* intron 49, and therefore, are not suitable to profile our compounds for their effect on the mHTT exon 1 mRNA species.

Human mHTT exon1 transcript (RNAseq data)

This is still an evolving field, but to date, there is no definitive demonstration that aberrant splicing of exon 1 fragment is the major cause of HD pathogenesis. Neueder et. al (2017) showed that the production of the *HTT* exon1 mRNA (generated by aberrant splicing) is dependent on longer CAG repeats. The *HTT* exon1 mRNA is detectable in postmortem brain samples from patients with juvenile CAG repeat lengths. In samples with repeat lengths in the adult-onset range, the levels of the *HTT* exon1 mRNA were comparable to levels in control brains. In addition, Yang et. al (2020) recently showed that the *HTT* mutant exon 1 is mainly generated from the canonical *HTT* mRNA, and not from the aberrant splicing of Exon1-Intron1. These splicing modifiers will target the full-length canonical *HTT* mRNA, and therefore, will prevent the formation of the any proteolytically cleaved exon 1 *HTT* protein fragment (from the full-length *HTT* protein).

Minor comments/suggestions:

1. The authors developed compound *HTT-D3* to achieve similar lowering levels in the periphery and in the brain by selecting for efficient *HTT* lowering together with a reduced *P-gp* efflux (lines 168-189). The writing of this section is clear, but readers may benefit from the addition of more context by describing the importance of *P-gp* efflux in designing drugs targeting the brain (e.g. explain that *P-gp* function is responsible for *MDR* in cancer, and that expression *P-gp* is high not only in the periphery, but also in brain capillary endothelial cells).

RESPONSE: We thank the reviewer for this important suggestion and have added the following sentence as well as supporting reference in 'Optimisation of HTT splicing modifiers' in 'Results' (lines 191–192) in the revised manuscript to provide more context.

“P-glycoprotein (P-gp) is one major transport protein expressed on BBB which limits entry of various drugs into the central nervous system.”

Reference added: International Transporter Consortium et al. “Membrane transporters in drug development.” *Nature reviews. Drug discovery* vol. 9,3 (2010): 215-36. doi:10.1038/nrd3028

2. The manuscript contains a large amount of data (and the authors have appropriately placed much of this data in supplementary figures). However, brevity required some very dense primary figures. Figure 1g,h show the plot of JEI following administration of vehicle or compound in several HTT introns, together with a schematic detailing the inclusion of the pseudo exon within intron 49 following treatment with the HTT-C1 compound. It would benefit the reader to separate these panels into a separate figure, and also include the schematic from Extended Data Fig. 9b.

RESPONSE: We thank the reviewer for this important suggestion and have made the suggested changes in the revised manuscript. We split Figure 1 into two figures (and added Fig. 2c from extended data Fig. 9b – see revised manuscript Figures)

3. Key data showing that the inclusion of the Stop Codon psiExon in intron 49 results in nonsense-mediated HTT mRNA decay (cycloheximide treatment and EP-PCR) is currently located in Extended Data Fig. 3. This data is important, and the authors should consider including it within a primary figure—for example, the new figure with the JEI plot. In the description of the cycloheximide experiment, the authors mention that total RNA was analyzed by error-prone PCR (EP-PCR) (described previously). A search of the manuscript did not reveal where this method was described previously, and a reference citation is needed.

RESPONSE: We agree with the reviewer and thank them for the suggestion. We have included the cycloheximide experiment in the main figures section (now Fig. 2f) in the revised manuscript. Regarding the error-prone PCR, we apologize for the mistake and thank the reviewer for catching the typing error. We have made the correction in 'Analysis of nonsense-mediated decay' in 'Methods' (line 402) in the revised manuscript. EP-PCR is Endpoint PCR here, the same as described elsewhere in the manuscript.

Reviewer #3 (Remarks to the Author):

Review for “Small molecule splicing modifiers with systemic HTT-lowering activity”

The authors report interesting findings about identification, characterization, and in vivo pharmacodynamics of novel splicing modulator compounds that drive downregulation of HTT mRNA and protein levels. From their novel high-throughput screening campaign, they identify two molecules, HTT-C1 and HTT-D1, that induced decreases in HTT protein levels in patient fibroblast cell lines. They further characterize these changes as driven by alterations to HTT mRNA splicing – notably the inclusion of a poison (pseudo)exon between Exons 49-50 of the HTT mRNA sequence. This results in the triggering of nonsense-mediated decay of the mRNA and decreased protein expression, due to inefficient translation of the transcript. Evaluation of the mechanism of action demonstrates that the splicing modulation is comparatively selective, resulting in altered splicing of several hundred transcripts (similar to the effects seen with SMN splicing modulators). The authors then demonstrate in vivo pharmacodynamic changes in HTT protein in the brains of transgenic mouse models of Huntington’s disease (HD). The work concludes with the indication that a clinical trial in HD patients is in the process of initiating, promising the potential for benefits to patients with a severe neurodegenerative disorder.

Overall, this work provides an encouraging portrait of the expanding potential for both targeting of mRNA splicing for therapeutic benefit as well as the prospect for treatment of a devastating disease with no currently approved therapeutic options. This work represents a strong leap forward in the field and should be considered for publication, following the authors addressing the issues listed below.

RESPONSE: We thank the reviewer and agree that our study provides an encouraging portrait of the expanding potential for both targeting of mRNA splicing for therapeutic benefit as well as the prospect for providing an oral treatment of HD. We thank the reviewer for their valuable comments.

Major Points:

1) *The authors introduce compounds HTT-C1 and HTT-D1 in figure 1, and note their discovery in a HTT protein high-throughput screen. HTT-C1 and D1, are highly structurally similar to branaplam and risdiplam, respectively. As such, the description of the HTT alternative splicing/exon inclusion would be a most likely MoA of these molecules. In addition, there are several features of the SMN2 and HTT splicing that are comparable (e.g., both molecules induce inclusion of an exon that is normally poorly spliced and the GA at the -2 and -1 positions of the 5’ss). There is an extensive amount of the manuscript that is dedicated to describing the fundamental MoA of these compounds, when the work could benefit from a comparative analysis of the selectivity (rather than a repeat of the published work from the branaplam and risdiplam manuscripts). As such, the manuscript would benefit from a comparison/contrast*

between the respective MoA of these two molecule classes, and their respective target mRNAs, relative to the selectivity of the target gene/splicing.

RESPONSE: We thank the reviewer for their comment and have addressed the comparison between the two classes of splicing modifiers (lines 127–138) in the revised manuscript. The comments below are addressed here: we (a) have added a table comparing HTT-C2 and SMN-C3 potencies (for HTT and SMN splicing) in Supplementary fig. 6, as well as amended Figure 4a and Supplementary table 2 to present the RNAseq. analysis of SMN-C3 (same class as HTT-D1 and risdiplam).

a. To that end, an evaluation of the SMN2 splicing changes induced by HTT-C1 and HTT-D1 – and, conversely, evaluating risdiplam-mediated splicing of the poison pseudoexon in HTT would be important to demonstrate.

b. Additionally, the authors report on several alternative splicing events induced by HTT-C1. There is no data shown for the alternative splicing induced by HTT-D1, this would be important to demonstrate a similar phenotype, especially as the molecules are from different chemotypes. This seq data could also be compared/contrasted with the results from the prior publications on risdiplam and branaplam, respectively to further bolster the selectivity of these molecules.

2) The authors show that – by RNAseq – there is an ~25% use of the poison pseudoexon junction reads, but show an ~80% decrease in protein level. How do they explain the discrepancy between a relatively minor depletion in the overall coding transcript for HTT (~75% full-length HTT – with no poison exon) and such a profound decrease in protein levels?

RESPONSE: The reviewer has asked an important question and we should clarify that the RNA-Seq detects and measures steady-state RNA level. The discrepancy between the relatively small splicing change and the larger gene/protein abundance changes reflects the disproportionate level of RNA degradation caused by the compound-triggered splicing change. The psiExon-inclusion transcript (HTT psiExon49a) is rapidly degraded and is not measured by RNA-Seq. Hence, the increased PSI of the psiExon underestimates the gene abundance decrease (from the RNA-Seq analysis).

3) Much of the work focuses on HTT protein (and mRNA) lowering, but there is little discussion of the mutant vs. wild-type isoforms. As the poison exon is quite downstream of the mutant allele site (in exon 1), the compounds would be expected to impact both alleles comparably.

RESPONSE: We thank the reviewer for this comment. As the reviewer pointed out, the compounds are expected to impact both wild type and mutant HTT alleles comparably. Our Supplementary Fig. 1e shows similar kinetics of HTT lowering – WT vs mHTT protein levels are

lowered comparably. We have added one sentence in ‘Optimisation of *HTT* splicing modifiers’ in ‘Results’ (lines 185–187) to further clarify the point in our revised manuscript.

a. Use of a mutant-specific HTT antibody (e.g. MW1), for example, in the HTT Hu97/Hu18 would enable detection of the lowering of the mutant allele protein levels. As mutant HTT is known to form comparatively stable higher-order protein aggregates, it would be important to assess the relative pharmacodynamics (PD) of mutant HTT lowering. This information would be relevant in the context of assessment pharmacokinetic (PK)/PD relationships.

RESPONSE: The reviewer asks a valid question and we have changed the % HTT remaining to % mHTT remaining in the revised manuscript and updated the text to clarify that in the two animal models mutant-selective HTT assays were used to assess mHTT lowering.

For both BACHD and Hu97/18 studies, quantitative protein detection assays were used to evaluate mHTT protein lowering in different tissues. One of the antibodies used in these assays is MW1 (Methods, ‘Electrochemiluminescence protein assay’, line 310). Also, for the CSF samples, a mutant-selective HTT assay was used (Methods, ‘Immunoprecipitation and flow cytometry mtHTT quantification’, lines 558–578), and the lowering correlated well with other tissues.

We also provided an example of a western blot (WB) to support similar mHTT lowering in BACHD mice. For this WB, we used AB2166, which detects both mHTT and murine HTT (served as an internal/loading control; Figure 5c).

For comparison, here we have added another WB (using MW1 antibody – see below) that demonstrates similar mHTT protein lowering in the BACHD brain with HTT-C2 (as observed with AB2166; Figure 5c).

b. Further relevant to the PK/PD relationship would be a comparison of the relative mRNA and protein changes over time in vivo. The authors present data for in vivo dosing, and show that protein level drop does not occur until several days post-treatment – reaching a nadir at 21 days. They show some recovery of protein after 5 days, restoring to baseline by 10 days post-removal of treatment. No data is shown for mRNA changes over time and this would be important to demonstrate the “target-proximal” PD changes, relative to PK.

RESPONSE: We agree with the reviewer and have added the RNA data for HTT-D3 in the Results section ('Optimisation of HTT splicing modifiers', lines 203–206) in the revised manuscript to demonstrate a strong RNA to protein correlation in the BACHD brain (Supplementary fig. 9d).

4) The Hu97/Hu18 mice have been shown to exhibit both pathologic and phenotypic alterations, due to the presence of the mutant HTT protein. This may be beyond the scope of the current manuscript, but the authors should consider presenting data from the mice as to the efficacy and tolerability of long-term treatment with HTT-D3. Treatment would be expected to reverse some of the phenotypes seen in these mice and evaluation of timing of treatment (e.g., juvenile vs. adult), relative to prevention vs. restoration of "wild-type" phenotypes (and their subsequent reversion to the "mutant" phenotypes) would be important.

RESPONSE: This is an important point, but as the reviewer mentioned, a long-term efficacy study is beyond the scope of this manuscript. The full-length HD mice used in this study display mild pathology and late onset phenotype progressing gradually over many months, with little signs of striatal degeneration. However, they offer an advantage over the partial fragment transgenic and knock-in models, especially when establishing target engagement and pharmacodynamic effects of our novel splicing modifiers that require the presence of a specific human HTT region.

Minor Points:

1) The authors note that, in mice with systemic depletion of HTT, there is evidence for pancreatitis. In the manuscript, they note that they switched the in vivo treatment paradigm from the HTT-C3 molecule to the HTT-D1 molecule, due to better partitioning between CNS and peripheral tissues, after an initial series of PK/PD experiments. Did the mice experience any pancreatic alterations due to a relatively transient lowering of HTT protein? This could be tracked by evaluation of amylase levels in serum, for example, and would be relevant for understanding of the relative onset of the phenotypes of HTT loss in CNS vs. peripheral tissues.

RESPONSE: We thank the reviewer for asking this question. The BACHD mice used for most of our in vivo studies have WT/murine HTT, and the splicing modifiers do not target murine HTT splicing. Therefore, in these animals only mutant HTT lowering was evaluated and the effect of WT HTT depletion in the CNS and/or periphery could not be assessed. The Hu97/18 mice were used to evaluate HTT-D3 (reduced P-gp efflux) instead of HTT-C2 (high efflux). This study was performed for 21 days and in this timeframe, we did not observe any gross organ pathology. To note, HTT was not completely depleted throughout the periphery even at the highest selected dose.

2) The authors note that the phenotype for these molecules is distinct from the SMN molecules (e.g., around line 212 in the discussion). Actually, the mechanism of action and phenotype is identical – both induce the inclusion of a poorly spliced exon. The difference is in the consequence of that exon inclusion. This is quite similar to the impact of alternative splicing for many splicing factors (e.g., HNRNP proteins) – as levels of these proteins rise, they induce alternative splicing of their own transcripts to trigger inclusion of poison exons to autoregulate the levels of their mRNA and protein. Some rephrasing around this section would help clarify this.

RESPONSE: As the reviewer correctly pointed out, the phenotype is identical – exon inclusion. We have rephrased our narrative in our Discussion section of the revised manuscript to clarify this point, by deleting a section drawing a comparison with risdiplam's MoA.

3) There are several minor questions around the alternative splicing:

a. Where the authors modified the sequence around the pseudoexon and the predicted ESE sequences, I wonder if their deletions and/or mutations altered any potential secondary structure that the RNA could form (e.g., stem loop) that could impede the access to U1.

RESPONSE: We appreciate the reviewer's comment and have addressed it here.

We carefully examined predicted secondary structures that could be created by the deletion and mutations and found no consistent structures that would cause occlusion of the 5' splice site (see below figure). For example, $\Delta 2$ left shows very similar secondary structure as the wild-type sequence. Other mutants examined seem to "open" the secondary structure around the 5'ss, which is inconsistent with the experimental evidence that they have less compound-activated splicing of the psiExon. Further investigation is ongoing to determine the exact nature of the role of the potential ESE sequences in the splicing of the psiExon.

RNA secondary structure prediction using RNAfold (<http://rna.tbi.univie.ac.at/cgi-bin/RNAWebSuite/RNAfold.cgi>)

b. Some comment could be made around the nature and consequences of the additional “off-mechanism” alternative splicing events seen in the RNAseq assessment. There was a summary statement about the number of exons and some comment about the presence of annotated and non-annotated exon events, but no further information was reported. A summary table with gene names, exon events, and predicted consequences for transcript & protein (e.g., does the new exon trigger frameshift & NMD, etc.) would be helpful.

RESPONSE: In supplementary table 2 and 3, columns “startSS_supp” and “endSS_supp” indicate how the splice sites are annotated. Columns “exon_i_inFrame” and “exon_i_stopCodon” indicate whether the psiExon will trigger frameshift or contain a stop codon. In supplementary table 1 (gene expression table), column “psiExon.coordinates” indicates the coordinates of psiExons as identified using RNA-Seq data of SH-SY5Y cells treated with HTT-C2 or HEK293 cells treated with U1-GA variant (from Supplementary table 3).

Finally, it is important to note that – although it has not been reported in peer-reviewed literature, Novartis has indicated that they are repositioning branaplam in a Phase 2a study in HD patients (<https://huntingtonsdiseasenews.com/2020/10/27/fda-gives-novartis-branaplam-orphan-drug-designation-for-huntingtons/>). From this, it is clear that the consideration of splicing modulation for treatment of HD is an idea that has gained acceptance.

RESPONSE: We agree with the reviewer and appreciate their comment – we believe this paper will be of great interest and significance in the HD field, as the first paper describing the identification of a novel compound-induced splicing event in the huntingtin pre-mRNA that

leads to pseudoexon inclusion and reduction in the burden of the key molecular determinant in HD pathology, the mutant HTT protein.

REVIEWERS' COMMENTS

Reviewer #2 (Remarks to the Author):

The authors have done an excellent job in responding to my questions and comments, and their edits have substantially improved the manuscript.

Reviewer #3 (Remarks to the Author):

The current format of the manuscript has answered most of the questions raised and is acceptable for publication.